# RetouchLLM: Training-free Code-based Image Retouching with Vision Language Models

## Abstract

Image retouching not only enhances visual quality but also serves as a means of expressing personal preferences and emotions. However, existing learning-based approaches require large-scale paired data and operate as black boxes, making the retouching process opaque and limiting their adaptability to handle diverse, user- or image-specific adjustments. In this work, we propose *RetouchLLM*, a training-free white-box image retouching system, which requires no training data and performs interpretable, code-based retouching directly on high-resolution images. Our framework progressively enhances the image in a manner similar to how humans perform multi-step retouching, allowing exploration of diverse adjustment paths. It comprises of two main modules: a visual critic that identifies differences between the input and reference images, and a code generator that produces executable codes. Experiments demonstrate that our approach generalizes well across diverse retouching styles, while natural language-based user interaction enables interpretable and controllable adjustments tailored to user intent.

## 1 Introduction

Image retouching is the process of enhancing the aesthetic visual quality of an image that suffers from photographic defects such as improper exposure, poor contrast, or color imbalance, typically through a sequence of global and/or region-specific adjustments. Image retouching plays a vital role in enhancing the visual quality of photographs and expressing personal emotions and aesthetics. The preferred retouching style varies significantly from person to person (Ouyang et al., 2023; Hu et al., 2018), and even for the same individual, it may differ depending on the subject or scene as different visual intentions arise (Wang et al., 2023). In response to this diversity, various automated deep learning-based image enhancement techniques (Duan et al., 2025; Ouyang et al., 2023; Wang et al., 2022; Kosugi, 2024; He et al., 2020) have been proposed. These methods typically involve training a model on image pairs, each consisting of an original and a retouched version in a desired style, allowing the model to learn and replicate the corresponding retouching patterns.

However, data-driven training approaches come with several notable limitations. They often require large-scale paired datasets for training, restricting the adaptability to new styles or environments, especially for general users without access to curated data. In addition, it is difficult to achieve fine-grained control that reflects specific user preferences or intentions, as the output tends to reflect an average learned from the training distribution (Wang et al., 2023). These models also function largely as black boxes, making it challenging to understand or intervene in the internal retouching process. Moreover, many of these methods apply edits to downscaled versions of the input image and later upscale the results, potentially degrading the original image quality.

In this work, we propose *RetouchLLM*, a training-free white-box image retouching system. Unlike data-driven approaches, our method (i) requires no training, (ii) provides transparent code-based retouching, and (iii) supports fine-grained adjustments through user instructions. Without relying on large-scale datasets of style-consistent paired images, RetouchLLM adapts flexibly to user-specific and image-specific preferences. A code-based design functions as a white-box, enabling the retouching process to be fully understandable and modifiable by users. By generating and executing code directly on high-resolution images without downscaling, it further enables high-fidelity enhancement suitable for real-world applications. In addition, the system accepts natural language instructions, enabling users to make personalized fine-grained edits in a controllable and interpretable manner.

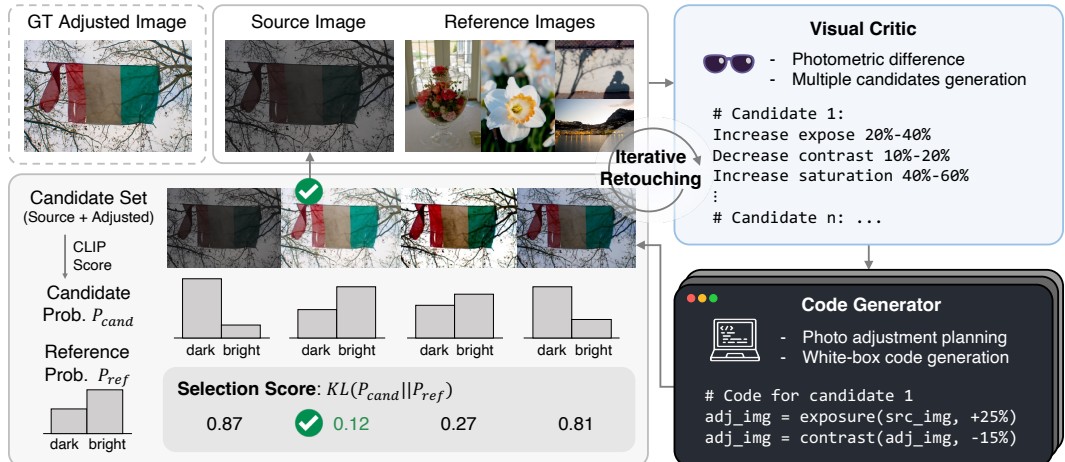

Figure 1: **Overview of our training-free white-box photo adjustment system.** Given a source image and style reference images, the visual critic gives multiple candidates of difference descriptions, and the code generator produces corresponding adjustment programs. The best candidate is selected according to the selection score, set as the new source, and the process iteratively continues until the stopping criterion is reached. The dashed box (GT Adjusted Image) is reference-only, outside the pipeline. Only dark/bright are shown for brevity, though eight prompts were used.

At its core, our approach uses an iterative retouching framework to progressively refine an image towards the target style rather than relying on one-shot edits. A selection score guides this process by reliably capturing style cues from reference images despite content differences, ensuring stable convergence and preventing drifting adjustments. RetouchLLM integrates two complementary modules: a visual critic, which identifies photometric differences and describes multiple candidate difference, and a code generator, which determines the editing sequence and produces executable retouching code. These components operate in a closed loop, enabling coarse global edits in early steps and fine local refinements later, thereby mirroring the natural workflow of human retouching.

We validate the effectiveness of RetouchLLM through both qualitative and quantitative experiments. Despite operating in a training-free manner, our model demonstrates favorable performance across various styles regardless of the backbone used. The results show that image quality improves progressively through iterative retouching. Ablation studies confirm that our selection score reliably captures style from reference images, even when the content differs, and both the visual critic and the code generator play essential roles in enhancing output quality. Moreover, we find that user interaction application via natural language instructions enables flexible and interpretable adjustments, allowing for fine-grained retouching. We summarize our key **contributions** as follows:

- We propose an iterative retouching framework guided by a style-guided selection score that ensures stable convergence toward the target style without requiring any training data.

- Our white-box, code-based design provides transparency and reproducibility, operating directly on high-resolution images and enabling reusable editing programs.

- By leveraging language-based models, our system supports interactive refinement via natural language instructions, enabling personalised retouching aligned with user intent.

## 2 RELATED WORK

**Automatic photo retouching.** Automatic photo retouching (Yan et al., 2016; Hu et al., 2018; Ke et al., 2022; Wu et al., 2024; Yang et al., 2024b; Duan et al., 2025; Tseng et al., 2022; Kim et al., 2020; Ouyang et al., 2023) has been extensively studied to automate the retouching pipeline by training deep models, allowing non-experts to attempt what was once an exclusive domain of professionals. These can be classified into two categories: Image-to-image translation methods and physics-based modeling methods. Translation methods train a model to generate a retouched image in the target style from an input image. Models such as U-Net (Ronneberger et al., 2015) and GANs (Goodfellow et al., 2014) are commonly used in this approach, but they often come with limitations in image

---

**Algorithm 1 RetouchLLM.** The algorithm iteratively adjusts the source image based on the given reference images. The process stops early (i) if the same image is selected for three consecutive steps, or (ii) if the generated descriptions suggest no additional edits are needed.

---

**Require:** Visual critic $f(\cdot)$, code generator $g(\cdot)$
**Input:** Initial source image $x_0^{\text{src}}$, reference set $\mathcal{Y} = \{y^j\}_{j=1}^M$, maximum iterations $T$, number of candidates $N$
**Output:** Adjusted image $x^*$
**for** $t = 0$ **to** $T - 1$ **do**
    Generate descriptions: $(d_1, \ldots, d_N) = f(x_t^{\text{src}}, \mathcal{Y})$
    **for** $i = 1$ **to** $N$ **do**
        Generate program: $g(d_i)$
        Adjust source image: $x_t^i = \text{Execute}(g(d_i), x_t^{\text{src}})$
    Construct candidate list: $\mathcal{C}_t = \left[ x_t^{\text{src}}, x_t^1, \ldots, x_t^N \right]$
    Best candidate selection: $i^* = \arg\min_{i \in \{0, \ldots, N\}} \text{SelectionScore}(\mathcal{C}_t[i], \mathcal{Y})$
    Update source image: $x_{t+1}^{\text{src}} \leftarrow \mathcal{C}_t[i^*]$
    **if** *stopping condition is met* **then**
        **break**
**return** $x^* = x_{t+1}^{src}$

---

resolution. These models also function as black-box, making it difficult to interpret the retouching process. Physics-based modeling treats the retouching pipeline as a combination of actual retouching filters and reframes the problem as estimating the appropriate filter combinations and their parameters. This makes it easier to design a more interpretable white-box system. However, most existing methods (Dutt et al., 2025; Duan et al., 2025; Kosugi, 2024) require training a model on thousands of paired images, which makes it difficult to incorporate new styles or filters without costly retraining. In contrast, our system needs no training and retouches an image directly from a few reference images of the desired style. Moreover, new filters can be flexibly incorporated without any additional retraining.

**LLMs and VLMs as model agents.** Large Language Models (LLMs) (Achiam et al., 2023; Chiang et al., 2023; Touvron et al., 2023a;b; Jiang et al., 2023; Abdin et al., 2024; Team et al., 2024b; Yang et al., 2024a) and Vision Language Models (VLMs) (Achiam et al., 2023; Dai et al., 2023; Liu et al., 2024; 2023; Team et al., 2024a; Wang et al., 2024; Hong et al., 2024a) are capable of performing a wide range of tasks, including question answering, information comparison and extraction, and generating text in various formats, such as natural language, JSON, code, and equations. By employing these models as agents, complex tasks like programming (Shinn et al., 2024; Surís et al., 2023; Gupta & Kembhavi, 2023; Hu et al., 2024), GUI understanding (Hong et al., 2024b), and decision making (Zhao et al., 2024), have been solved in a more organized and structured manner.

However, prior work (Dutt et al., 2025; Kosugi, 2024) utilizing LLMs or VLMs does not incorporate any iterative feedback mechanism or support agent-like collaboration with the user, thereby limiting flexibility, personalization, and transparency in the process. In contrast, our approach is training-free, white-box, and supports interactive user collaboration. Specifically, we utilize a VLM as an agent to infer the photographic differences, and an LLM as an agent to generate a retouching process in Python code.

## 3 RETOUCHLLM: TRAINING-FREE WHITE-BOX IMAGE RETOUCHING

Despite the success of existing retouching approaches, their reliance on style-specific training limits their adaptability to new domains or settings. Moreover, the black-box nature of some prior methods makes it difficult to interpret, intervene in, or modify the retouching process. In this work, we propose *RetouchLLM*, a training-free, white-box photo retouching pipeline that iteratively adjusts images, as illustrated in Fig. 1 and summarized in Algorithm 1. We do so using a language-based foundation model as a *visual critic* and a *code generator*. Our approach generalizes to a wide range of domains, and provides transparency and controllability by generating interpretable retouching program.

In Sec. 3.1, we describe how our method performs iterative image retouching, introduce the selection score, and analyze the convergence behavior. In Secs. 3.2 and 3.3, we provide explanations for each module. More details of the RetouchLLM, such as prompts, can be found in Sec. B of Appendix.

## 3.1 ITERATIVE RETOUCHING

Iterative image retouching is performed by integrating the visual critic and code generator modules. At each iteration $t$, a source image $x_t^{\text{src}}$ and style reference images $\mathcal{Y} = \{y^j\}_{j=1}^M$ are passed to the visual critic, which produces $N$ difference descriptions. Given each description, the code generator plans how to adjust the image accordingly and generates executable program to perform the retouching. As a result, $N$ programs are generated and executed, producing $N$ retouched images $x_t^1, ..., x_t^N$. By exploring multiple adjustment paths in parallel, the system avoids overcommitting to a single direction, ensuring that potentially more optimal adjustment strategies are considered.

**Selection score.** At each iteration $t$, we select one retouched image from the candidate set $\mathcal{C}_t$, which consists of $N + 1$ images: $N$ retouched images $\{x_t^i\}_{i=1}^N$ and one source image $x_t^{\text{src}}$. We include the source image in the set $\mathcal{C}_t$ to prevent degenerate updates. To select one appropriately, we propose a SelectionScore, which identifies the most suitable image while avoiding unnecessary computational cost. Our SelectionScore is formulated based on CLIP space alignment (Radford et al., 2021). Since the contents of retouched images and reference images differ, we focus on extracting retouching style using filter-related text prompts. We use $K$ text prompts $\{z_1, ..., z_K\}$, constructed as pairs of contrasting prompts for $K/2$ filters, where $K$ is even. Although we employ seven filters in total, text prompts are constructed only for the four global filters, yielding $K = 8$ prompts in total.

With CLIP, let the image encoder be $\phi_{\text{img}}(\cdot) : \mathcal{X} \to \mathbb{R}^D$ and the text encoder be $\psi_{\text{text}}(\cdot) : \mathcal{T} \to \mathbb{R}^D$. The image embedding is $\mathbf{v}(y) = \phi_{\text{img}}(y)$, and the text embedding is $\mathbf{e}_k = \psi_{\text{text}}(z_k)$. For each candidate image $x_t \in \mathcal{C}_t$ in iteration $t$, and reference image $y \in \mathcal{Y}$, we compute their logits with respect to the $K$ text embeddings as

$$\ell_k(x_t) = \langle \mathbf{v}(x_t), \mathbf{e}_k \rangle, \qquad \ell_k(y) = \langle \mathbf{v}(y), \mathbf{e}_k \rangle. \tag{1}$$

These logits can be converted to probabilities:

$$P_{cand} = p_k(x_t) = \frac{\exp(\ell_k(x_t))}{\sum_{r=1}^K \exp(\ell_r(x_t))}, \qquad q_k(y) = \frac{\exp(\ell_k(y))}{\sum_{r=1}^K \exp(\ell_r(y))}. \tag{2}$$

Then, we define the SelectionScore $\sigma : \mathcal{X} \to \mathbb{R}_{\geq 0}$ used by Algorithm 1 as

$$\sigma(x_t, Y) = D_{\text{KL}}(p(x_t) \| \bar{q}) = \sum_{k=1}^K p_k(x_t)\big(\log p_k(x_t) - \log \bar{q}_k\big), \tag{3}$$

where $P_{ref} = \bar{q}_k = \frac{1}{M} \sum_{j=1}^M q_k(y^j)$ summarizes all of the reference images with respect to the text prompts. Finally, the selected image in iteration $t$ is obtained by

$$x_{t+1}^{src} \leftarrow \mathcal{C}_t[i^*], \quad \text{where } i^* = \arg\min_{i \in \{0,...,N\}} \sigma(\mathcal{C}_t[i], \mathcal{Y}), \tag{4}$$

with $\mathcal{C}_t = [x_t^{src}, x_t^1, \ldots, x_t^N]$ being the candidate set for the current iteration $t$.

**Role of the source image in convergence.** As iterations progress, the difference between the source and reference images decreases, and the visual critic reports fewer differences. This dynamic feedback enables the code generator to adjust its planning, focusing solely on the remaining elements that require adjustment. By always including the current source image in the candidate list, the process ensures that the selected image cannot be strictly worse than in the previous iteration.

Formally, let $\sigma_t = \sigma(x_t, \mathcal{Y})$ denote the selection score at iteration $t$.

$$\sigma_{t+1} = \sigma(x_{t+1}, \mathcal{Y}) \leq \sigma(x_t, \mathcal{Y}), \tag{5}$$

showing that the sequence $(\sigma_t)$ is nonincreasing. Since

$$\sigma_t = D_{\text{KL}}(p(x_t) \| \bar{q}) \geq 0, \tag{6}$$

the sequence $(\sigma_t)$ is bounded below by zero. By the properties of bounded monotone sequences, $(\sigma_t)$ therefore converges to some finite limit. Although this convergence does not imply global optimality (since the candidate set $\mathcal{C}_t$ may only partially cover the search space), we observe that further improvements beyond 10 iterations are small. We therefore fix the iteration budget to $T = 10$.

In summary, the iterative process continues until one of two stopping conditions holds: (1) the visual critic reports no significant differences across all retouching elements, or (2) a predefined maximum number of $T = 10$ iterations is reached.

## 3.2 VISUAL CRITIC: DESCRIBING PHOTOMETRIC DIFFERENCES

We aim to predict the photometric differences of each filter between a source image and style reference images in a training-free manner. Traditionally, such differences have been predicted by training models for each style, which incurs the cost of re-collecting thousands of paired images and retraining the model whenever a new style is needed for image adjustment. Moreover, the preferred retouching style varies from person to person and even between images, further increasing the need for personalized models. To address these problems, we employ a vision–language model as a visual critic to understand diverse images and identify their differences.

Nevertheless, accurately identifying the photometric differences remains a challenge, even for humans. Furthermore, there is often no single correct answer since preferred adjustments may vary significantly between individuals depending on their subjective taste and intent. To mitigate this ambiguity, we generate multiple candidate difference descriptions and propagate them for further exploration. At each iteration $t$, we produce N descriptions $\{d_t^i\}_{i=1}^N$. This strategy increases the chance of including a valid description. If the probability of generating a suboptimal description in the single-candidate case is $p$, then the probability of success, *i.e.*, having at least one useful description, is $P(\text{success}) = 1 - p^N$, where $P(\text{all suboptimal}) = p^N$. Thus, increasing $N$ substantially improves the likelihood of capturing a valid adjustment direction, providing robustness against error accumulation.

To empirically support this intuition, we conduct a toy experiment on brightness range prediction in Table 1 (see Sec. C.1 of Appendix for experimental details). The probability that the search space contains a correct direction is 24.3% with a single prediction and 71.3% with two candidates. This simplified test highlights the effectiveness of multi-candidate generation, supporting our theoretical analysis and motivating its use in our iterative image retouching setting. Once the visual critic is unable to describe the photometric differences, the retouched image will be close to the reference images and the retouching process will be complete (after stopping condition is met).

Table 1: **Photo adjustment brightness range prediction.**

| Model | Correct |
|---|---|
| Random | 16.7 |
| GPT-5 (Single) | 24.3 |
| GTP-5 (Multi) | 71.3 |

## 3.3 CODE GENERATOR: PLANNING AND IMPLEMENTING

Given the difference descriptions $d$, we perform actual image retouching by generating the program $g(d)$. However, this task introduces two significant challenges: the interdependencies among retouching elements, and the computational burden caused by high-resolution images. For example, DSLR images often have extremely high resolution, making it computationally expensive for the model to directly modify pixel values at full resolution (Bakhtiarnia et al., 2024). To address these issues, we leverage the planning and executable code generation capabilities of a large language model.

The retouching program $g(d)$ is expressed as a composition of 7 photometric operations (exposure, contrast, saturation, temperature, highlight, shadow, texture), but the framework can be readily extended to a larger set. The photometric operation pool is

$$\mathcal{P}_\theta = \{\text{exposure}(\theta_{\text{exp}}), \dots, \text{texture}(\theta_{\text{tex}})\}. \tag{7}$$

More formally, for each photometric description, the code generator selects a subset of filters $h$, their ordering, and computes their arguments $\theta$, and applies them sequentially to the source image.

$$h = (h_1, h_2, \dots) \subset Perm_t(\mathcal{P}_\theta) \quad \rightarrow \quad x_t^i = (h_1 \circ h_2 \circ \dots)(x_t^{src}), \tag{8}$$

where $Perm_t(\cdot)$ describes permuting the order of filter operations for time-step $t$. Each operation in $\mathcal{P}_\theta$ can be applied to images of any size without additional processing. Thus, the procedure is resolution-independent. In addition, unlike black-box models that map an input image directly to its retouched image, our approach is fully white-box: the generated program $h$ explicitly specifies all operations and parameters, ensuring interpretability, editability, and reproducibility.

## 4 EXPERIMENTS

### 4.1 EXPERIMENTAL SETUP

**Datasets.** We evaluate our method using two publicly available datasets: MIT-Adobe FiveK (Bychkovsky et al., 2011) and PPR10K (Liang et al., 2021). While traditional methods need training, our

Table 2: **Retouching performance across multiple styles.** Gray text indicates models fine-tuned on the target style using reference images. Black text denotes zero-shot models evaluated without any task-specific training. Results for RetouchLLM are reported using the GPT-5 implementation.

| Style | Method | MIT-Adobe FiveK | | | | PPR10K | | | |
|-------|--------|-----------|-----------|-----------|-----------|-----------|-----------|-----------|-----------|
| | | PSNR(↑) | SSIM(↑) | LPIPS(↓) | ΔE(↓) | PSNR(↑) | SSIM(↑) | LPIPS(↓) | ΔE(↓) |
| 1 | RSFNet | 18.03 | 0.773 | 0.178 | 18.34 | 17.81 | **0.819** | **0.106** | 19.77 |
| | PG-IA-NILUT | 20.54 | 0.743 | 0.168 | 12.13 | **19.60** | 0.674 | 0.146 | 15.24 |
| | Z-STAR | 16.01 | 0.607 | 0.397 | 17.70 | 16.13 | 0.662 | 0.325 | 19.63 |
| | RetouchLLM | **21.68** | **0.900** | **0.072** | **10.89** | 19.31 | 0.817 | 0.150 | **14.95** |
| 2 | RSFNet | 17.89 | 0.754 | 0.188 | 18.09 | **21.93** | **0.870** | **0.079** | 13.21 |
| | PG-IA-NILUT | 18.04 | 0.632 | 0.221 | 17.46 | 21.42 | 0.760 | 0.104 | **11.82** |
| | Z-STAR | 16.23 | 0.592 | 0.412 | 19.39 | 16.89 | 0.629 | 0.336 | 16.62 |
| | RetouchLLM | **21.13** | **0.867** | **0.094** | **12.20** | 20.91 | 0.837 | 0.116 | 12.07 |
| 3 | RSFNet | 17.60 | 0.780 | 0.162 | 17.36 | 21.27 | 0.834 | **0.072** | 13.50 |
| | PG-IA-NILUT | 19.48 | 0.686 | 0.232 | 14.35 | 21.25 | 0.710 | 0.112 | 12.59 |
| | Z-STAR | 15.40 | 0.597 | 0.379 | 20.54 | 18.38 | 0.679 | 0.321 | 15.50 |
| | RetouchLLM | **21.32** | **0.897** | **0.081** | **12.17** | **21.49** | **0.853** | **0.105** | **12.27** |

approach does not require a training phase. Thus, we only utilize the test pairs for evaluation. To evaluate our RetouchLLM, we construct source-reference image pairs. The pairs reflect a common user behavior in real-world retouching workflows, where users often refer to similar content images as references, *e.g.*, refer to a green nature image when retouching a mountain scene. To mimic this behavior, we utilize CLIP (Radford et al., 2021) to extract image-level logits and compute the pairwise KL divergence across the dataset. Reference images are selected based on their similarity.

**Training-free baseline.** Since existing image retouching methods typically require training, we adopt Z-STAR (Deng et al., 2024), a zero-shot style transfer model, as our training-free baseline. Z-STAR represents the content and style images through dual denoising paths in the latent space and guides the denoising process of the content image using the style latent codes via cross-attention reweighting. We use the source image to be retouched as the content image and the reference image as the style image, and treat the resulting output as the training-free baseline results.

**Implementation details.** We use five reference images per sample ($M = 5$), and the visual critic generates three candidate descriptions per iteration ($N = 3$) by default. To improve fine-grained retouching stability, we provide the visual critic with image-level statistics, *e.g.*, pixel mean, std, etc, in the prompt. We implement the visual critic and code generator using four LLMs: GPT-5 (OpenAI, 2025), Gemini-1.5-Pro (Team et al., 2024a), Qwen2.5-VL-72B (Bai et al., 2025), and InternVL3-14B (Zhu et al., 2025). Further implementation details are provided in Sec. C of Appendix.

**Metrics.** Following previous work (Ke et al., 2022; Wang et al., 2022; Wu et al., 2024), we utilize Peak Signal-to-Noise Ratio (PSNR), Structural Similarity Index Measure (SSIM), Learned Perceptual Image Patch Similarity (LPIPS), and $\Delta E$, which represents the $\mathcal{L}_2$ distance in the CIELAB color space. Higher PSNR and SSIM values, with lower LPIPS and $\Delta E$ values, indicate better performance.

## 4.2 EXPERIMENTAL RESULTS

**Retouching performance across diverse styles.** We evaluate RetouchLLM on diverse retouching styles against (1) Z-STAR, a training-free retouching baseline, and (2) supervised models RSFNet (Ouyang et al., 2023) and PG-IA-NILUT (Kosugi, 2024), each fine-tuned to the target style using reference images. Details of fine-tuned models are provided in Sec. C.2 of the Appendix.

We present the results in Table 2. Our RetouchLLM significantly outperforms Z-STAR in all metrics and styles. Furthermore, it outperforms the two fine-tuning methods in all metrics in MIT-Adobe FiveK, while reaching competitive results in PPR10K, demonstrating the adaptability of RetouchLLM across a wide range of retouching styles. In addition, Table 3 highlights the extensibility of Retouch-LLM to different model backbones, con-

Table 3: **Retouching performance across multiple models.** All values are averaged over eight retouching styles: five from Adobe FiveK and three from PPR10K.

| Model | PSNR(↑) | SSIM(↑) | LPIPS(↓) | ΔE(↓) |
|-------|---------|---------|----------|-------|
| RSFNet | 18.69 | 0.798 | 0.144 | 16.82 |
| PG-IA-NILUT | 19.73 | 0.692 | 0.173 | 14.05 |
| Z-STAR | 16.28 | 0.623 | 0.368 | 18.30 |
| RetouchLLM | | | | |
| w/ GPT-5 | 20.75 | 0.858 | 0.101 | 12.76 |
| w/ Gemini-1.5-Pro | 20.41 | 0.857 | 0.102 | 13.03 |
| w/ Qwen2.5-VL-72B | 20.00 | 0.844 | 0.103 | 13.68 |
| w/ InternVL3-14B | 20.72 | 0.857 | 0.106 | 12.75 |

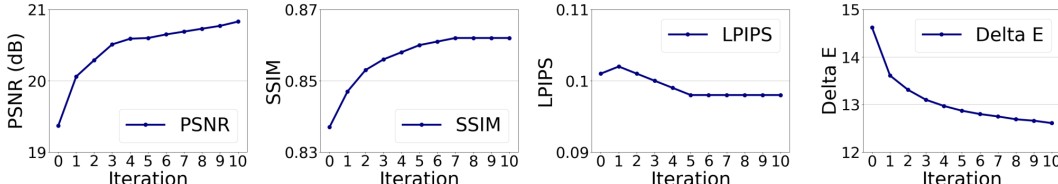

Figure 2: **Quantitative results over 10 iterations.** All metrics show consistent improvement over iterations. Higher PSNR and SSIM, and lower LPIPS and ΔE, indicate closer similarity to the GT.

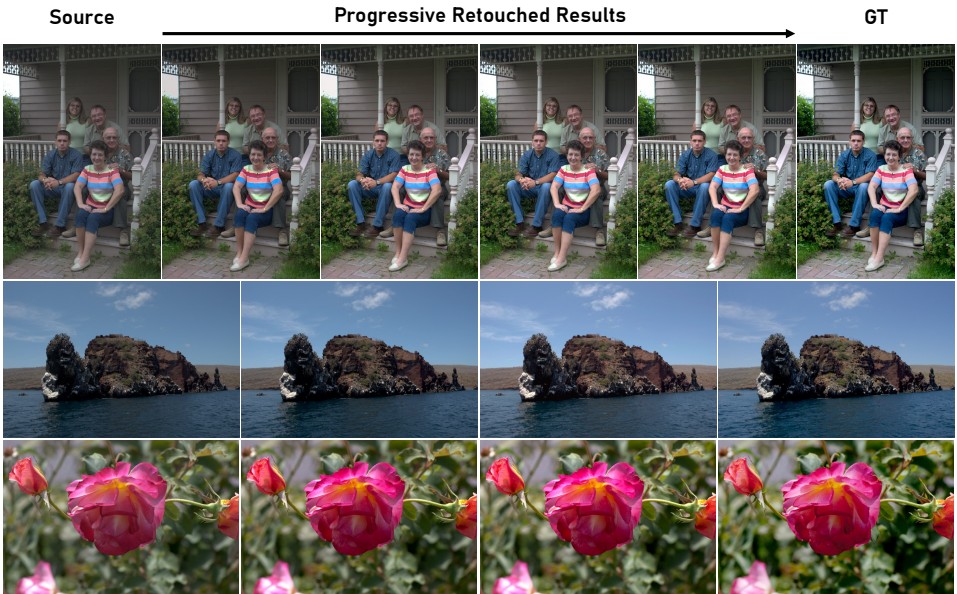

Figure 3: **Qualitative results of progressively retouched images.** In each row, the leftmost image is the source, the rightmost is the GT, and the middle images show the progressively retouched results.

sistently yielding superior performance over existing methods regardless of the backbone used. The style-wise full results of Table 2 and Table 3 are given in Sec. A.1 of the Appendix.

**Iterative retouching.** To evaluate the effectiveness of our iterative retouching framework, we conduct experiments with the GPT-5 version of RetouchLLM. The quantitative results in Fig. 2 report the metric-wise trend averaged over all 8 retouching styles, revealing consistent improvements as the image is progressively retouched. The most significant changes occur in the early iterations, while the gains diminish in later steps, suggesting convergence. This behavior indicates that RetouchLLM performs image retouching in a coarse-to-fine manner, making significant global adjustments in the beginning and gradually refining finer details as the iterations proceed. The qualitative results in Fig. 3, starting from the source image on the left, show a progressive enhancement toward the target style, as verified by comparison with the ground truth (GT) image on the right. Notably, all retouching operations are performed directly on the original high-resolution images without any resizing, preserving fine details throughout the iterative process. The qualitative results of high-resolution images are provided in Figs. 7, 8, and 9 of the Appendix.

**Plausibility assessment.** We evaluate our method in the plausibility assessment setting (Dutt et al., 2025), where each input image is associated with five expert retouchings and the model output is scored against the closest among them. This protocol evaluates how well

Table 4: **Score consistency (mean ± standard deviation) under a similar-content reference setting.**

| Method | PSNR(↑) | SSIM(↑) | LPIPS(↓) | ΔE(↓) |
|---|---|---|---|---|
| Exposure (Hu et al., 2018) | 15.12 | 0.63 | 0.14 | - |
| Unpaired (Kosugi & Yamasaki, 2020) | 21.73 | 0.83 | 0.12 | - |
| RSFNet (Ouyang et al., 2023) | 21.85 | 0.88 | 0.08 | - |
| InstructP2P (Brooks et al., 2023) | 16.99 | 0.61 | 0.22 | - |
| MGIE (Fu et al., 2024) | 22.94 | 0.74 | 0.08 | - |
| MonetGPT (Dutt et al., 2025) | 23.75 | 0.90 | **0.07** | - |
| RetouchLLM | **25.48** | **0.92** | 0.09 | **7.57** |

Table 5: **Ablation of RetouchLLM modules.** Z-STAR is the baseline for training-free retouching. (a) The code generator produces the retouching code based on the statistics of images instead of using a textual description. (b) The visual critic directly generates codes. (c) Our RetouchLLM. In the paired setup, the ground truth retouched image corresponding to the source is available. The unpaired setup is a more general case, where the reference images have different contents but a desirable style.

| | Visual Critic | Code Generator | Paired | | | | Unpaired | | | |
|---|---|---|---|---|---|---|---|---|---|---|
| | | | PSNR(↑) | SSIM(↑) | LPIPS(↓) | $\Delta$E(↓) | PSNR(↑) | SSIM(↑) | LPIPS(↓) | $\Delta$E(↓) |
| | Z-STAR | | 20.01 | 0.732 | 0.196 | 13.60 | 16.29 | 0.595 | 0.399 | 18.99 |
| (a) | ✗ | ✓ | 26.78 | 0.947 | **0.050** | 6.69 | 19.74 | 0.866 | 0.096 | 13.78 |
| (b) | ✓ | ✗ | 27.58 | **0.959** | 0.052 | 5.79 | 20.71 | 0.863 | 0.092 | 12.81 |
| (c) | ✓ | ✓ | **29.21** | 0.956 | 0.053 | **5.23** | **22.19** | **0.909** | **0.070** | **10.07** |

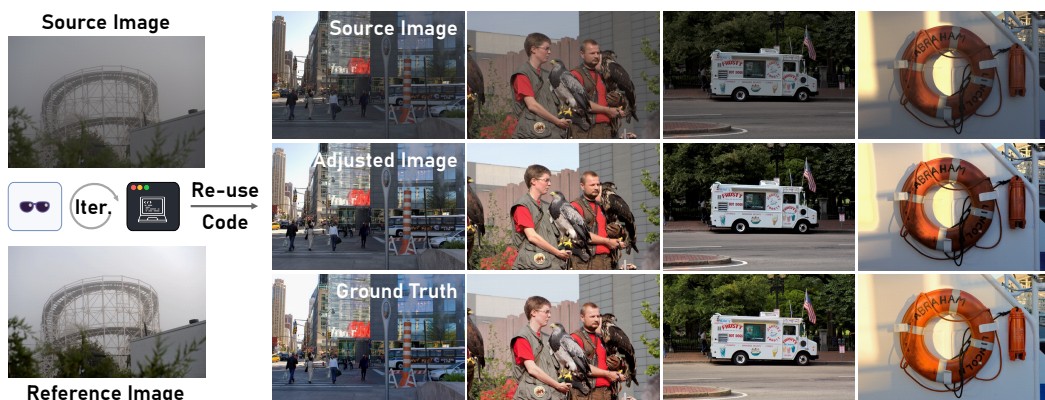

Figure 4: **Applying the restored filter.** The paired setup enables extracting a more faithful and reusable retouching code that can be applied to other images like a preset filter.

a model can produce edits that fall within the range of human editing preferences. Following this protocol, we evaluate our method by randomly sampling 400 images from the 500 test images in MIT-Adobe FiveK (Bychkovsky et al., 2011) and reporting the corresponding scores. Table 4 shows that our method achieves strong performance, suggesting that its outputs are not only quantitatively superior to other methods but also more closely aligned with the diversity of expert-level adjustments.

**Ablation and comparison of selection scores.** In Table 6, we evaluate the effectiveness of the proposed selection score introduced in Sec. 3.1 against alternative methods: (1) RGB-channel histograms, (2) YUV-channel histograms, (3) Gram matrix similarity commonly used in style transfer, (4) our KL CLIP score using prompts re-

Table 6: **Effectiveness of the proposed selection score.** Results are reported using InternVL3 on FiveK style 1.

| | Selection Score | PSNR(↑) | SSIM(↑) | LPIPS(↓) | $\Delta$E(↓) |
|---|---|---|---|---|---|
| (1) | RGB hist. | 21.62 | 0.906 | 0.078 | 11.51 |
| (2) | YUV hist. | 20.94 | 0.894 | 0.078 | 11.66 |
| (3) | Gram matrix | 21.49 | 0.906 | 0.072 | 11.15 |
| (4) | KL CLIP all | 21.93 | 0.899 | 0.072 | 10.58 |
| (5) | KL CLIP global | **22.19** | **0.909** | **0.070** | **10.07** |

garding all filters including local ones, and (5) our default KL CLIP score using only global filters. Details of each score are given in Sec. C.3 of the Appendix. The result shows that our proposed method (global) achieves the best performance, demonstrating its ability to reliably capture style characteristics from reference images even when their content differs from the source image.

**Ablation of RetouchLLM modules.** In Table 5, we conduct an ablation study on MIT-Adobe FiveK style 1 using InternVL3, systematically modifying or removing modules. We design two ablation variants: (a) we remove the visual critic and instead feed image-level statistics (*e.g.*, mean, standard deviation) directly to the code generator, which provides insight about how the form of image information representation affects retouching performance; (b) we merge the visual critic and code generator into a single VLM that directly generates retouching code by comparing the source and reference images, without explicitly describing their differences, which examines the role of natural language as a semantic textual bottleneck in guiding the image retouching process.

Table 7: **Ablation of the number of candidates.** (Default: $N = 3$)

| # Cand. | PSNR(↑) | SSIM(↑) | LPIPS(↓) | ΔE(↓) |
|---|---|---|---|---|
| 1 | 21.27 | 0.889 | 0.078 | 10.91 |
| 3 | 22.19 | 0.909 | 0.070 | 10.07 |
| 5 | **22.76** | **0.914** | **0.069** | **9.74** |

Table 8: **Ablation of the number of style reference images.** (Default: $M = 5$)

| # Ref. | PSNR(↑) | SSIM(↑) | LPIPS(↓) | ΔE(↓) |
|---|---|---|---|---|
| 1 | 20.35 | 0.890 | 0.080 | 11.37 |
| 3 | 21.80 | 0.904 | **0.070** | 10.13 |
| 5 | **22.19** | **0.909** | 0.070 | **10.07** |

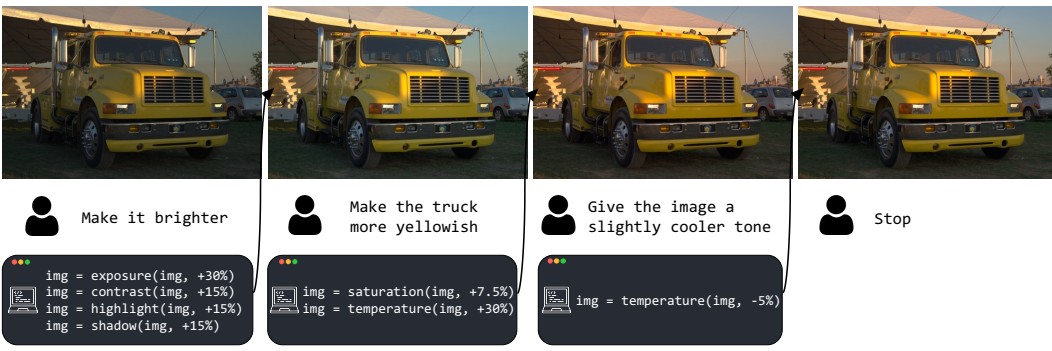

Figure 5: **User interactive retouching.** The user gives instructions to retouch images towards the desired style. These retouched images can then be fed back into the pipeline for further retouching.

We compare variants under paired and unpaired setups. In the paired setup, the target image paired with the source serves as a reference, allowing direct validation of the generated code against expert retouching. In the unpaired setup, which we take as the default and more general setting, reference images with different content but similar style are used, requiring the model to generalize its retouching logic. In Table 5, all variants outperform the baseline Z-STAR. RetouchLLM (c) achieves the best performance across all metrics in the unpaired setup, and in the paired setup it also attains the highest PSNR and $\Delta E$, with the values of other metrics that are nearly indistinguishable from the top-performing variants. This suggest that both components contribute to retouching quality. The strong performance in the paired setup further suggests that a training-free approach can effectively leverage prior knowledge to handle fine-grained photo retouching. In addition, Fig. 4 illustrates a practical case of the paired setup, showing that the generated program can be reused to achieve a similar style.

**Ablation of design choices.** We conduct ablation studies on two key system design choices: the number of candidates and the number of reference images. In Table 7, as the number of candidates increases, the performance improves, which is consistent with the explanation in Sec. 3.2 and the results of Table 1. However, since a larger number also increases computational cost, we use three candidates in practice. Table 8 shows that increasing the number of reference images improves performance, with the best results obtained using five, as the model benefits from richer stylistic cues.

**User study.** We conducted a user study to assess perceptual preferences across methods. We collected responses from 40 participants over 30 samples (1,200 responses in total). We ensured that the participants cover different genders, nationalities, and come from geographically different regions. More details are in Appendix C.4. The results show a strong preference for our method: NILUT: 16.42%, RSFNet: 9.67%, Z-STAR: 3.17%, and Ours: 70.75%. This indicates that users consistently favored the outputs of RetouchLLM over the other baselines in terms of matching the target style.

**Robustness to changes in reference images.** Users generally rely on reference images with similar content. To evaluate the score consistency under such conditions, we randomly select five images from the top ten candi-

Table 9: **Score consistency (mean ± standard deviation) under a similar-content reference setting.**

| PSNR(↑) | SSIM(↑) | LPIPS(↓) | ΔE(↓) |
|---|---|---|---|
| 22.34± 1.55 | 0.918± 0.016 | 0.068± 0.011 | 9.44± 1.04 |

dates produced by CLIP-based retrieval and use each of them as a reference for retouching. This procedure is repeated seven times, and we report the trimmed mean and standard deviation by excluding the maximum and minimum values. The evaluation set is identical to that used in Table 4 for the

ablation study. Table 9 shows that the standard deviation remains small, indicating that the method behaves reliably when the reference images contain different but semantically similar content.

However, when the reference images contain very different content, the performance may decrease because current VLMs can struggle to consistently perceive and abstract the same style across heterogeneous scenes. Incorporating additional adaptation using retouching data to improve robustness in such challenging cases would be a promising direction for future work.

### 4.3 APPLICATION: USER INTERACTION

We further demonstrate an application where RetouchLLM is adapted from reference-based retouching to user-interactive retouching. In this setting, reference images are replaced with natural language instructions, and the automatic selection score is replaced with explicit user choices. This design allows the system to more directly reflect user preferences and supports an interactive workflow in which the retouching process can be iteratively refined through simple language commands.

The qualitative results of Fig. 5 illustrate how RetouchLLM progressively improves image quality while reflecting the user's intended style. We observe that the model effectively retrieves and composes relevant filters based on user instructions, while also adjusting their intensity in a controlled and interpretable manner. For instance, in response to an instruction such as "make the truck more yellowish," the model increases both saturation and color temperature to enhance the yellow tone. When subsequently asked to "give the image a slightly cooler tone," it reduces the temperature by a relatively small amount (*e.g.*, 5%). These results demonstrate that the model can faithfully interpret and execute user-provided natural language instructions for personalized retouching. More user interactive retouching examples can be found in Figs. 17 and 18 in the Sec. A.4 of the Appendix.

## 5 CONCLUSION

In this work, we present RetouchLLM, a training-free white-box system for interactive image retouching. By integrating an iterative refinement framework with a style-guided selection score, our approach achieves stable convergence without the need for paired training data. Extensive experiments demonstrate that it generalizes well across diverse styles and supports high-resolution editing with transparent, code-based operations. Beyond quantitative improvements, the ability to follow natural language instructions enables personalized and user-aligned retouching. For future work, we plan to extend our system with a broader set of editing filters and operations, enabling richer adjustment paths beyond the current set of retouching tools. Another important direction is evaluating human–AI interaction in practical workflows, studying how users issue natural language instructions and how effectively the system adapts to their preferences. We believe these directions will push interactive retouching toward more practical, personalized, and trustworthy real-world applications.

### ETHICS STATEMENT

This work includes a user study to evaluate the quality of generated results. We collected responses from 40 participants. Participants were volunteers who provided informed consent. We ensured diversity in gender, nationality, and geographic region among the participants. No personally identifiable information was collected, and no compensation was provided. The study posed no foreseeable risk of harm and was conducted in accordance with the ICLR Code of Ethics.

### REPRODUCIBILITY STATEMENT

We take several steps to ensure the reproducibility of our work. Detailed experimental settings, dataset descriptions, and implementation details are included in the main paper and appendix. The core algorithm is provided in the main paper for clarity. In addition, we provide the source code and instructions as part of the supplementary materials to facilitate replication of our results.

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

In this Appendix, we include additional qualitative and quantitative results, method details, and experimental details, which are not included in the main paper.

# A   ADDITIONAL RESULTS

## A.1   RETOUCHING ACROSS MULTIPLE STYLES

We complement Table 2 and Table 3 in the main paper by providing evaluations across all retouching styles in Table 10. We evaluate RetouchLLM on eight diverse retouching styles, including five styles from the MIT-Adobe FiveK dataset and three from the PPR10K dataset. For each style, we randomly select 30 source images for testing, and each source image is paired with five reference images. Unlike conventional source–GT pairs that share the same content but differ in style, our setting uses source–reference style pairs, where the content may differ and the goal is to adjust the source image to match the reference style. As described in the main text, we construct these pairs using CLIP (Radford et al., 2021), where we extract image-level logits and compute pairwise KL divergence across the dataset. The list of test images will be released along with the code.

RetouchLLM retouches images without any task-specific training. We compare our method against (1) Z-STAR (Deng et al., 2024), a training-free retouching baseline, and (2) RSFNet (Ouyang et al., 2023) and PG-IA-NILUT (Kosugi, 2024), two supervised models fine-tuned for each style using the corresponding reference images. The results in Table 10 demonstrate the adaptability of RetouchLLM across a wide range of retouching styles, and extensibility to different model backbones. In addition, we provide a qualitative comparison in Figure 6, where Z-STAR produces distorted results due to its diffusion-based style transfer mechanism, while our method yields results most similar to the ground truth compared to supervised methods.

## A.2   HIGH-RESOLUTION IMAGE RETOUCHING RESULTS

RetouchLLM retouches the image based on Python code; thus, it can operate independently of the image resolution. Figures 7, 8, and 9 present the retouching results on high-resolution images from MIT-Adobe FiveK (Bychkovsky et al., 2011). RetouchLLM infers the target style from five reference images and then applies iterative code-based adjustments to retouch the source image. The results demonstrate that RetouchLLM effectively extracts the photometric style attributes from the reference images and applies them to high-resolution content without degradation, confirming its capability for resolution-agnostic and content-preserving retouching.

## A.3   APPLYING RESTORED FILTER

Figure 16 presents additional examples demonstrating the practical utility of the paired setup introduced in Fig. 4 of the main paper. Given a single pair of a source image and a corresponding target image, our RetouchLLM extracts a retouching program that can be reused as a preset filter. This enables consistent adjustments across multiple photos taken under similar conditions (*e.g.*, the same scene or session). The extracted program can also be applied to new images independent of image resolution, as long as their starting point and intended target style are comparable, demonstrating the flexibility and scalability of our approach.

## A.4   USER INTERACTION RESULTS

Our default pipeline takes a source image and a set of reference images as input. Using a visual critic and a code generator, RetouchLLM iteratively produces adjusted image candidates. Among these, the most style-consistent image is selected via a score-based selection mechanism and then used as the source image for the next iteration. This iterative process continues until the final result is obtained.

In contrast, the user-interaction mode replaces the reference images with user instructions as input. Instead of automated score-based selection, the user can manually select the preferred result at each iteration, allowing RetouchLLM to adapt to the preferences of the individual user. This interactive loop continues until the user is satisfied with the result. This application is made possible by our language model–based modules and iterative design, which together allow the system to flexibly interpret user guidance.

Table 10: **Retouching performance across multiple styles (full results of Table 2 in the main paper).** Gray text indicates models fine-tuned on the target style using reference images. Black text denotes zero-shot models evaluated without any task-specific training.

| Style | Method | PSNR(↑) | SSIM(↑) | LPIPS(↓) | ΔE(↓) |
|---|---|---|---|---|---|
| FiveK A | RSFNet | 17.55 | 0.747 | 0.204 | 17.34 |
| | PG-IA-NILUT | 20.00 | 0.713 | 0.189 | 13.46 |
| | Z-STAR | 16.61 | 0.622 | 0.383 | 16.77 |
| | RetouchLLM (GPT-5) | 21.24 | 0.860 | 0.086 | 12.43 |
| | RetouchLLM (Gemini-1.5-Pro) | 21.08 | 0.862 | **0.082** | 12.27 |
| | RetouchLLM (Qwen2.5-VL) | 20.04 | 0.856 | 0.084 | 12.69 |
| | RetouchLLM (InternVL3) | **21.55** | **0.871** | 0.090 | **11.95** |
| FiveK B | RSFNet | 18.03 | 0.773 | 0.178 | 18.34 |
| | PG-IA-NILUT | 20.54 | 0.743 | 0.168 | 12.13 |
| | Z-STAR | 16.01 | 0.607 | 0.397 | 17.70 |
| | RetouchLLM (GPT-5) | 21.68 | 0.900 | 0.072 | 10.89 |
| | RetouchLLM (Gemini-1.5-Pro) | 21.55 | 0.905 | 0.074 | 10.99 |
| | RetouchLLM (Qwen2.5-VL) | 21.39 | 0.889 | 0.075 | 11.42 |
| | RetouchLLM (InternVL3) | **22.19** | **0.909** | **0.070** | **10.07** |
| FiveK C | RSFNet | 17.89 | 0.754 | 0.188 | 18.09 |
| | PG-IA-NILUT | 18.04 | 0.632 | 0.221 | 17.46 |
| | Z-STAR | 16.23 | 0.592 | 0.412 | 19.39 |
| | RetouchLLM (GPT-5) | **21.13** | 0.867 | 0.094 | **12.20** |
| | RetouchLLM (Gemini-1.5-Pro) | 21.06 | **0.872** | **0.092** | 12.44 |
| | RetouchLLM (Qwen2.5-VL) | 20.24 | 0.843 | 0.101 | 13.95 |
| | RetouchLLM (InternVL3) | 20.69 | 0.871 | 0.097 | 12.46 |
| FiveK D | RSFNet | 17.40 | 0.765 | 0.160 | 16.97 |
| | PG-IA-NILUT | 17.68 | 0.615 | 0.224 | **14.95** |
| | Z-STAR | 14.60 | 0.593 | 0.391 | 20.27 |
| | RetouchLLM (GPT-5) | **18.93** | **0.834** | **0.105** | 15.08 |
| | RetouchLLM (Gemini-1.5-Pro) | 17.94 | 0.809 | 0.109 | 16.16 |
| | RetouchLLM (Qwen2.5-VL) | 17.33 | 0.818 | 0.112 | 16.32 |
| | RetouchLLM (InternVL3) | 18.30 | 0.812 | 0.114 | 15.95 |
| FiveK E | RSFNet | 17.60 | 0.780 | 0.162 | 17.36 |
| | PG-IA-NILUT | 19.28 | 0.687 | 0.222 | 14.79 |
| | Z-STAR | 15.40 | 0.597 | 0.379 | 20.54 |
| | RetouchLLM (GPT-5) | **21.32** | **0.897** | **0.081** | 12.17 |
| | RetouchLLM (Gemini-1.5-Pro) | 20.22 | 0.882 | 0.084 | 13.22 |
| | RetouchLLM (Qwen2.5-VL) | 19.06 | 0.864 | 0.092 | 14.71 |
| | RetouchLLM (InternVL3) | 21.24 | **0.897** | 0.089 | **11.64** |
| PPR10K A | RSFNet | 17.81 | **0.819** | **0.106** | 19.77 |
| | PG-IA-NILUT | **19.60** | 0.674 | 0.146 | 15.24 |
| | Z-STAR | 16.13 | 0.662 | 0.325 | 19.63 |
| | RetouchLLM (GPT-5) | 19.31 | 0.817 | 0.150 | 14.95 |
| | RetouchLLM (Gemini-1.5-Pro) | 19.62 | 0.828 | 0.144 | **14.67** |
| | RetouchLLM (Qwen2.5-VL) | 18.98 | 0.804 | 0.148 | 16.15 |
| | RetouchLLM (InternVL3) | 19.13 | 0.807 | 0.156 | 15.23 |
| PPR10K B | RSFNet | 21.93 | **0.870** | **0.079** | 13.21 |
| | PG-IA-NILUT | 21.42 | 0.760 | 0.104 | **11.82** |
| | Z-STAR | 16.89 | 0.629 | 0.336 | 16.62 |
| | RetouchLLM (GPT-5) | 20.91 | 0.837 | 0.116 | 12.07 |
| | RetouchLLM (Gemini-1.5-Pro) | 20.82 | 0.855 | 0.117 | 12.10 |
| | RetouchLLM (Qwen2.5-VL) | 21.17 | 0.845 | 0.113 | 12.26 |
| | RetouchLLM (InternVL3) | 21.19 | 0.842 | 0.119 | 12.03 |
| PPR10K C | RSFNet | 21.27 | **0.872** | **0.072** | 13.50 |
| | PG-IA-NILUT | 21.25 | 0.710 | 0.112 | 12.59 |
| | Z-STAR | 18.38 | 0.679 | 0.321 | 15.50 |
| | RetouchLLM (GPT-5) | 21.49 | 0.853 | 0.105 | **12.27** |
| | RetouchLLM (Gemini-1.5-Pro) | 20.96 | 0.842 | 0.116 | 12.39 |
| | RetouchLLM (Qwen2.5-VL) | 21.78 | 0.831 | 0.101 | 11.93 |
| | RetouchLLM (InternVL3) | **21.50** | 0.845 | 0.111 | 12.71 |

From the second iteration, we incorporate retouching history to improve the precision of the model's interpretation of the user's instructions. Specifically, the visual critic receives the adjustment history, represented as image statistics, from previous iterations. This helps narrow down the ambiguity in the user instructions by providing contextual cues based on prior adjustments. The detailed qualitative process of user interaction is illustrated in Fig. 17, and additional qualitative results are in Fig. 18.

| Ground Truth | RSFNet | PG-IA-NILUT | Z-STAR | Ours |
| --- | --- | --- | --- | --- |

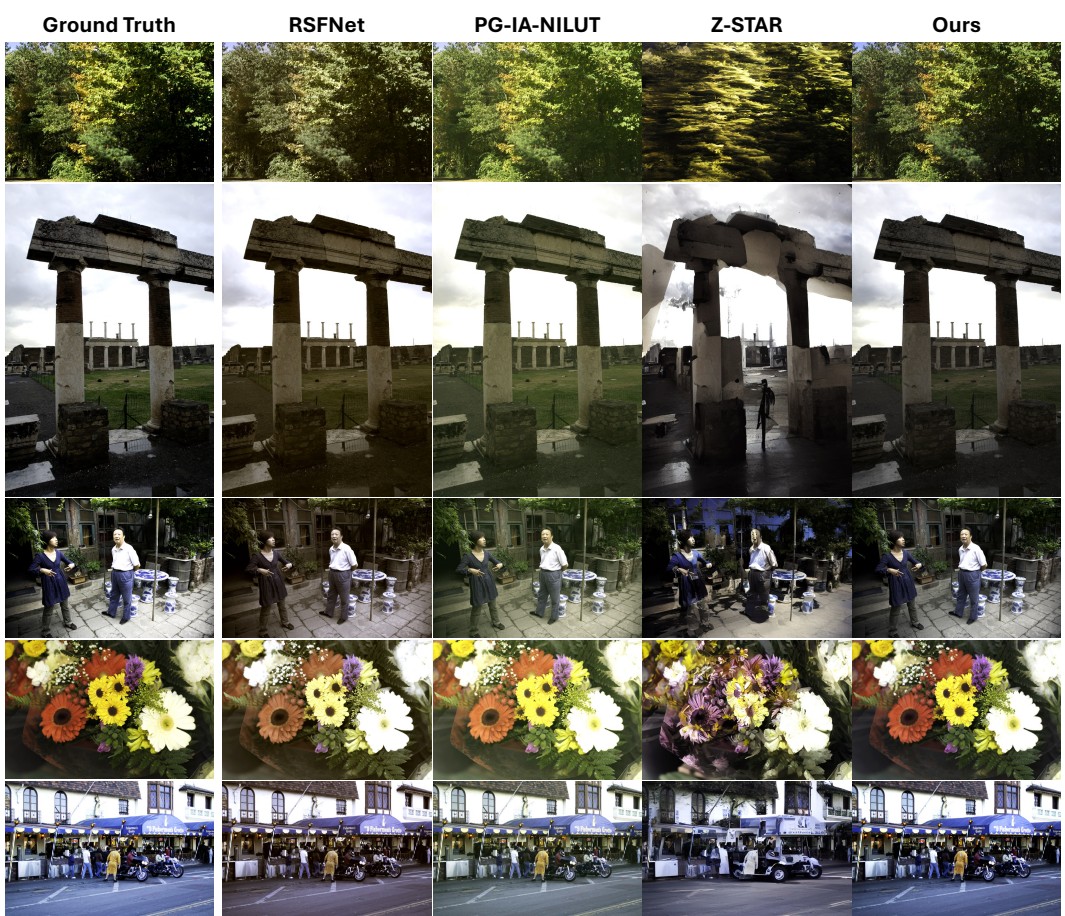

Figure 6: **Qualitative comparison results**

In Fig. 17, the user intends to make the source image significantly brighter with a warmer tone. Since RetouchLLM cannot determine the precise degree of adjustment in a single step, it generates three candidate images using different adjustment strategies. Specifically, the exposure is increased by approximately 30% to 80%, and the color temperature is raised by 15% to 50%, respectively. This range of candidates allows the model to explore diverse interpretations of the user's instruction. After the user selects the most preferred image among the three candidates, the selected image is used as the source for the next iteration of editing based on a new instruction from the user. This process is repeated iteratively: in each iteration, RetouchLLM generates three new candidates, and the user selects one to proceed. For simplicity, the three candidate outputs and user selections are omitted in Fig. 6 of the main paper and Fig. 18 of the supplementary material. Instead, only the final selected outputs for each iteration are shown to better illustrate the progressive retouching process.

In the bottom example of Fig. 18, the model reduces the saturation by 30% in the first iteration. In the second round, the user provides a vague instruction "reduce the saturation further", which the model interprets as requiring a stronger adjustment relative to the previous one and applies an 80% reduction. In the third iteration, the user says "slightly lower the saturation". Compared to the previous 80% change, the model interprets the given instruction as a much smaller adjustment, reducing saturation by 29%. This example demonstrates how the system leverages the retouching history to interpret ambiguous instructions more precisely. By referencing the magnitude and context of previous adjustments, the model can better infer relative terms such as "further" or "slightly", allowing for more user-aligned and consistent retouching behavior across iterations.

## A.5 GENERATED DESCRIPTIONS AND CORRESPONDING CODES SAMPLES

RetouchLLM performs image retouching using difference descriptions from the visual critic and retouching code generated by the code generator. Figures 20, 21, and 22 show examples of actual

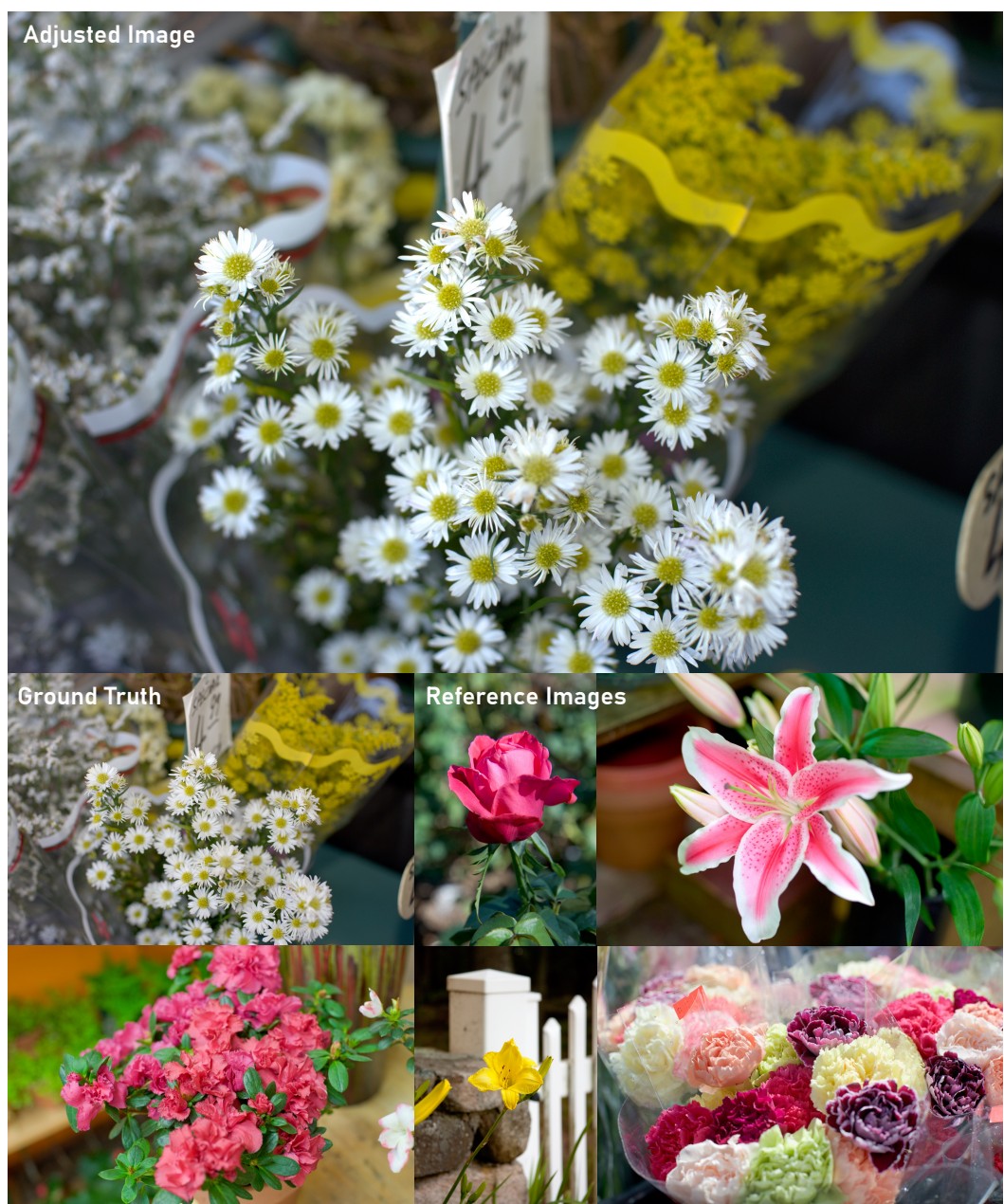

Figure 7: **High resolution qualitative samples 1**

outputs produced by both components. The visual critic generates three candidates of difference descriptions, each of which describes the difference across all filters. Based on each description, the code generator produces the corresponding retouching code.

In particular, rather than adjusting all components simultaneously, RetouchLLM first applies a global brightness adjustment, followed by local brightness, color tone, and texture adjustments. Since the effect of a given adjustment can depend on the sequence of operations, retouching workflows typically begin with global edits such as brightness or exposure before proceeding to more fine-grained adjustments. Our model incorporates this knowledge and plans a process for each iteration based on both the difference description and general editing conventions. As illustrated in Fig. 19, our method selectively applies adjustments across iterations and progressively refines the image, resembling an interactive human workflow.

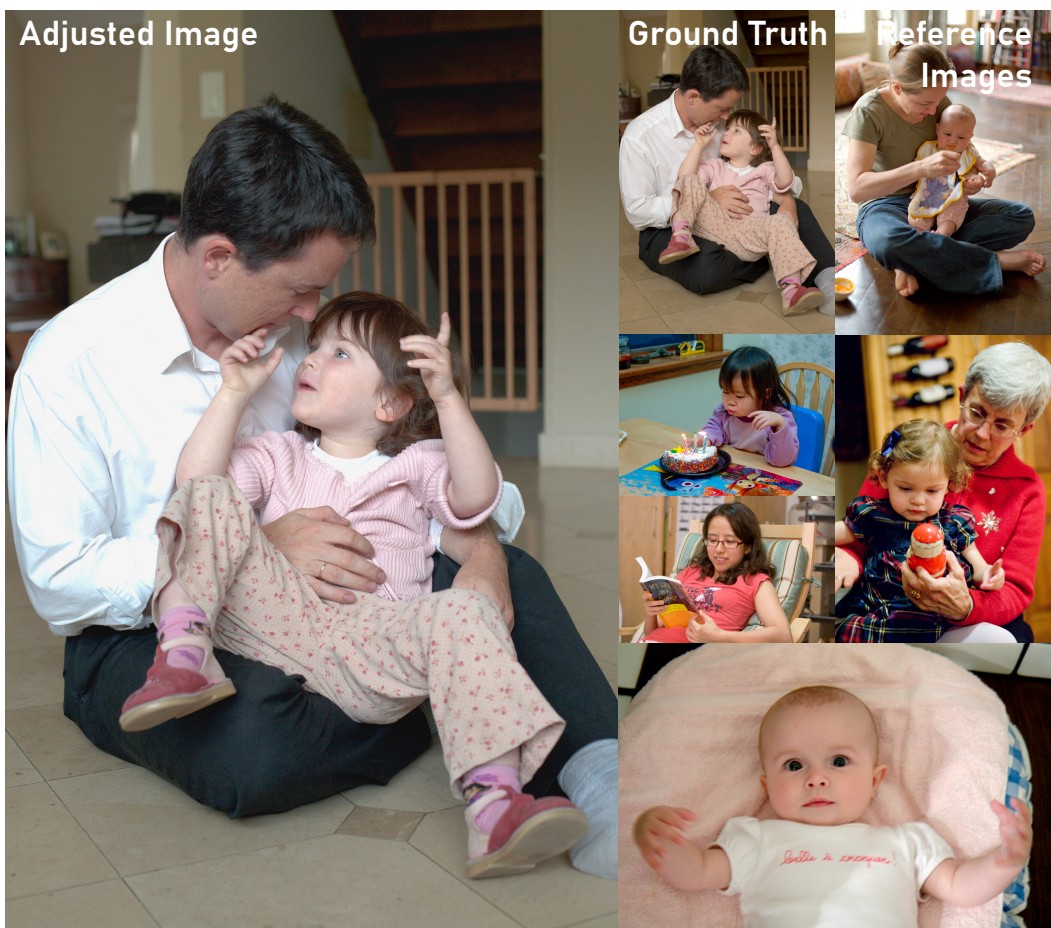

Figure 8: **High resolution qualitative samples 2**

## A.6 FAILURE CASES

While our system performs robustly in most cases, we do observe several types of failure cases during the process. First, the visual critic occasionally omits the decision for further editing. If any per-filter analysis is present but the final decision omits further editing, we proceed as if additional retouching is still required. Second, the code generator may occasionally output code that is not directly executable. For example, including placeholder comments such as "`source_image = ... # assume source_img is already defined`" can lead to execution errors. In such cases, we re-query the model with a different temperature, allowing up to three attempts. If all three retries fail, the system skips the current retouching and proceeds to the next iteration. In practice, these cases are not very common at all and are often resolved by the subsequent iteration. Both failure cases stem from the dependency on the existing external modules of VLMs and LLMs. As these modules improve, the failure cases of the overall proposed system will be reduced.

Figure 10 illustrates failure cases observed in the final results. In the first row, although the dark stone tomb is successfully brightened so that its boundaries become clearly visible, this also leads to an over-exposed sky region. In the second row, while the sky is stylized well according to the target style, the floor area becomes slightly less yellow than desired. These errors are expected to be alleviated once mask-based local editing is incorporated in the future. The last row shows cases where the overall adjustment does not perfectly match the ground truth, resulting in either insufficient brightness or a slightly cool tone. Since our method infers the style from a reference image that does not share the same content, rather than learning from paired data, the result is not identical to the ground truth. Nevertheless, it still produces plausible adjustments compared to the original image.

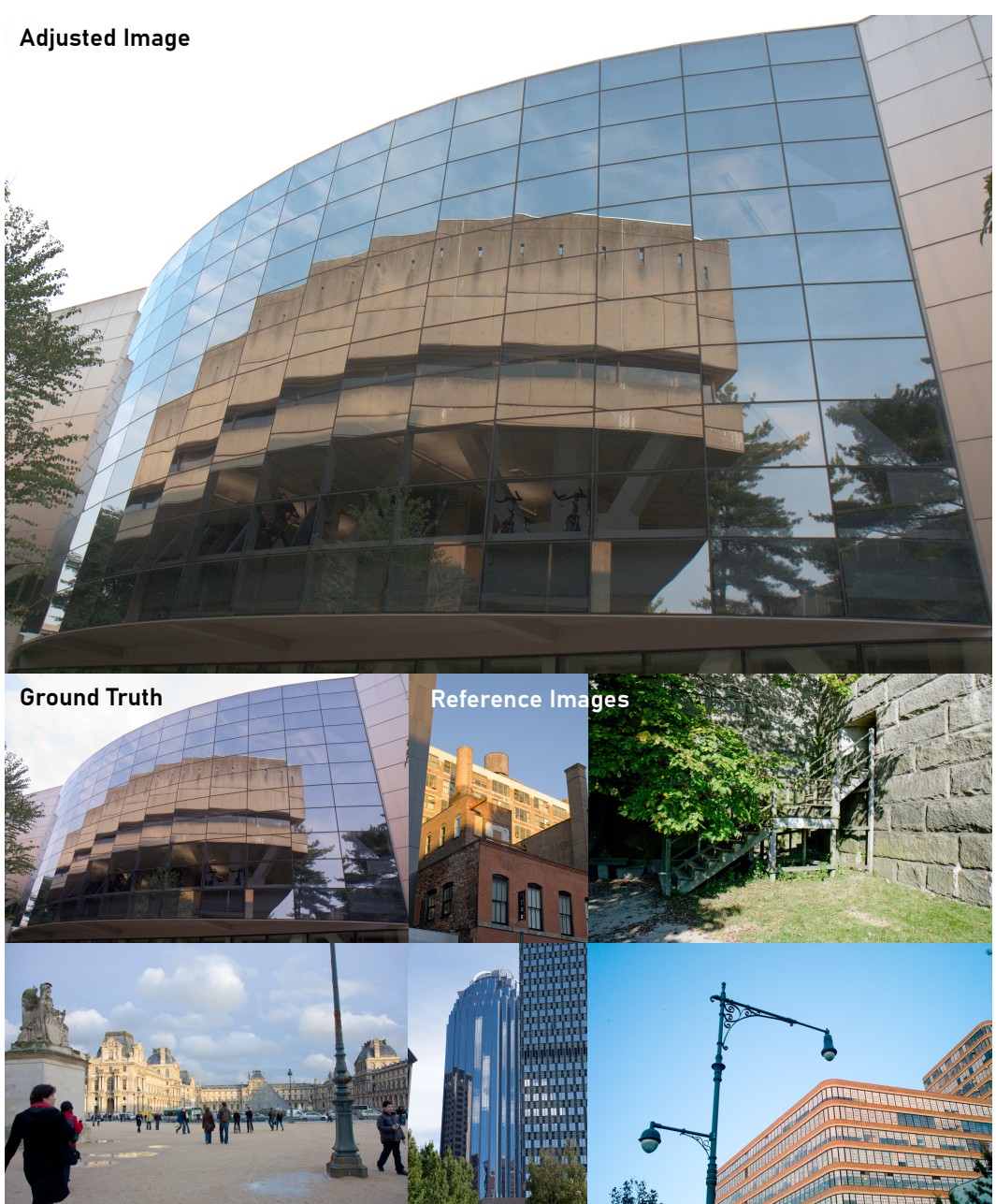

Figure 9: **High resolution qualitative samples 3**

## A.7 APPLICABILITY FOR LOCAL IMAGE EDITING

To evaluate whether our pipeline can be extended toward localized retouching, we additionally incorporated local editing using segmentation-based masking. Specifically, a target region mask was obtained via SAM (Kirillov et al., 2023) and applied after the original global enhancement. The results in Fig. 11 demonstrate that local editing further improves similarity to the ground truth compared to using global operations alone, *e.g.*, the first sample's PSNR is increased from 23.79 to 29.33, indicating that the framework can be extended beyond purely global corrections by integrating mask-guided refinements.

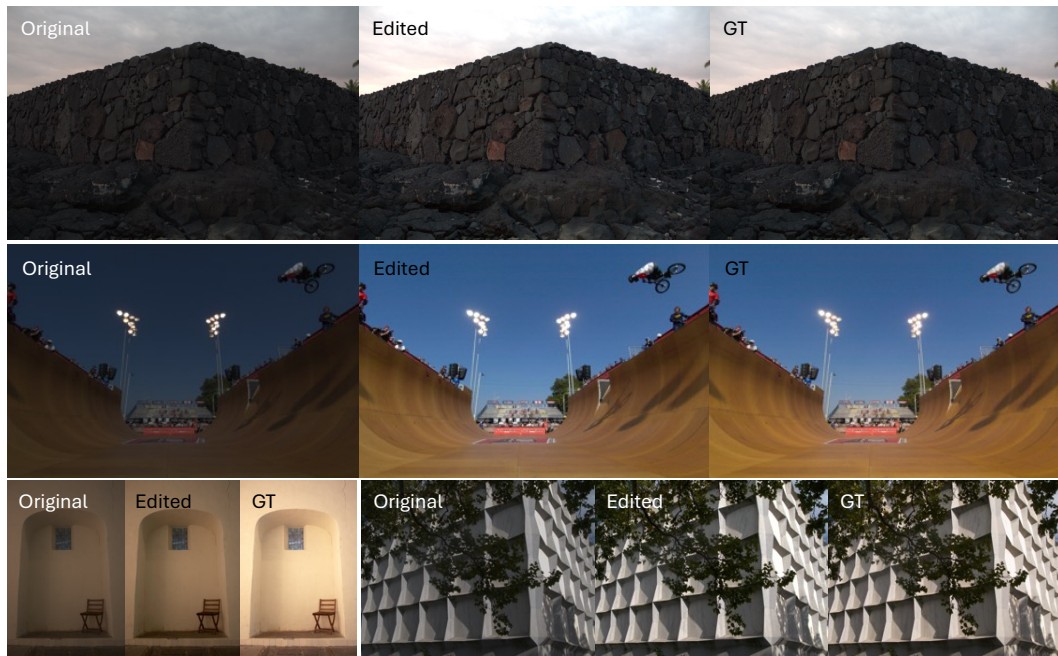

Figure 10: **Failure cases.** (Top) the sky area is overexposed. (Middle) the floor area looks less yellow. (Bottom left) the overall exposure is a bit low. (Bottom right) the colors lean slightly cool.

## B  DETAILS OF RETOUCHLLM

### B.1  PIPELINE

We summarize the full procedure of RetouchLLM in Algorithm 1. RetouchLLM employs a visual critic based on a vision-language model (VLM) and a code generator based on a large language model (LLM). Notably, neither model has been explicitly trained for the image retouching task. The system performs iterative retouching until either the maximum number of iterations $T$ is reached or the early stopping conditions are met. We define two stopping conditions: (1) Score-based early stopping, and (2) stop signal from the visual critic. First, if the source image is selected as the best candidate for three consecutive iterations, we assume that no further improvement is necessary. Second, if the visual critic explicitly includes a "stop" in the overall component of its difference description, the process is terminated early. During the iteration, the source image of the current iteration is included in the selection candidate set to ensure the reversibility in case the model outputs an unsatisfactory result. In the first iteration, a rule-based adjusted image is included as a warm-start image in the selection candidate set.

When the visual critic produces a description, we include statistics of the given images as well as those of the source and reference images, to enable a more fine-grained and quantitative explanation of inter-image differences. To support fine-grained analysis, we computed a set of image statistics: pixel-level mean, median, and standard deviation; top and bottom 10% intensities; RGB channel-wise means; Laplacian variance (for sharpness); saturation mean, standard deviation, minimum, and maximum; and the mean values of the L and b channels.

RetouchLLM retouches an image in under two minutes without any fine-tuning, whereas both RSFNet and PG-IA-NILUT require a very large corpus of training data and more than two days of training (using the authors' provided code). Fine-tuning on a smaller subset reduces this cost but severely degrades performance, as both models are prone to overfitting when trained on a few examples.

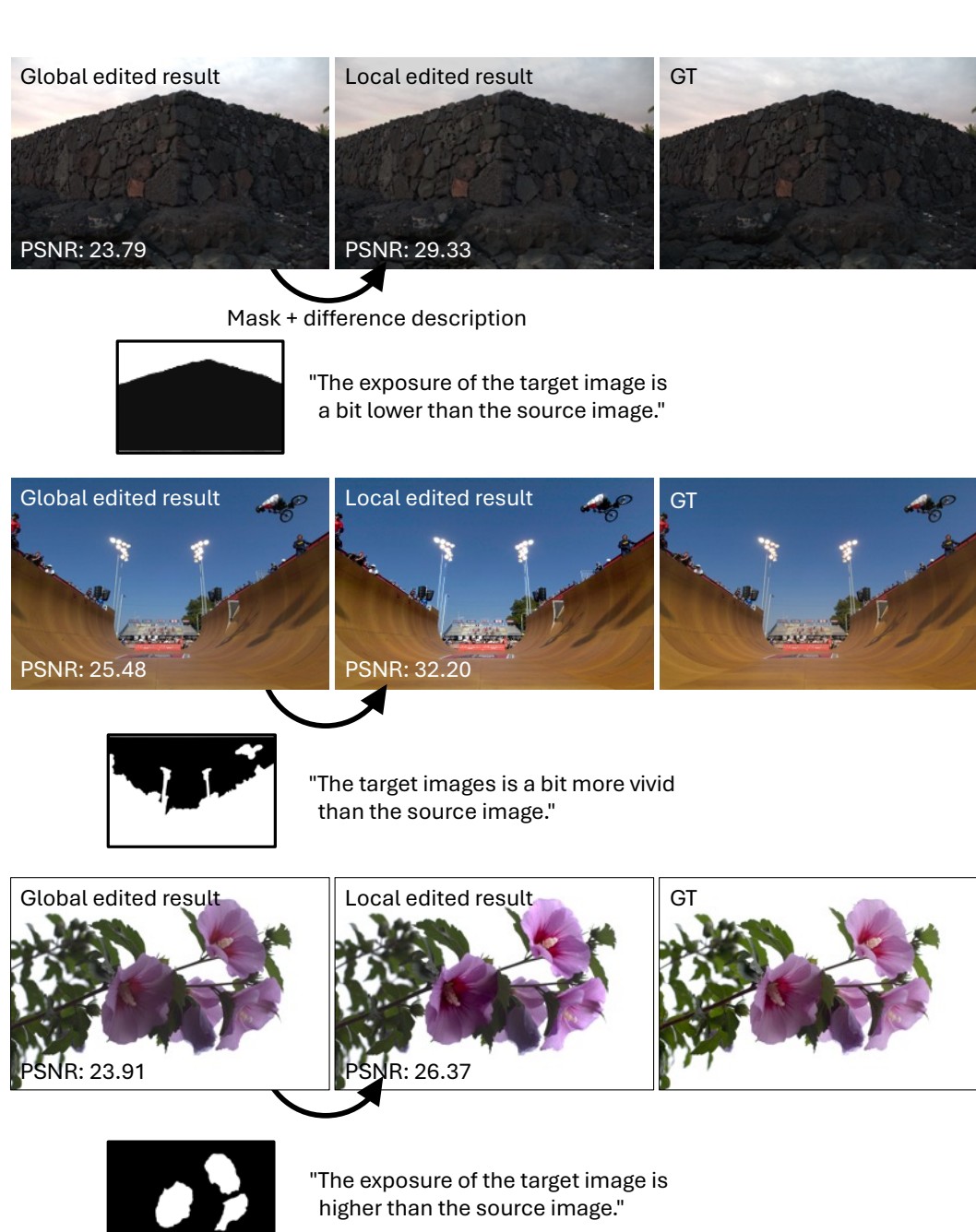

Figure 11: **Local image editing.** Local editing with a segmentation mask further improves similarity to the ground truth compared to global operations alone, demonstrating the possibility of extending our framework toward localized retouching. Note that the local region represented with white on the mask is edited, and the mask is predicted by SAM (Kirillov et al., 2023).

## B.2 PROMPT

We provide the exact prompt used in our system below. The system prompt for the visual critic is in 12, the user prompt for the visual critic in 13, the system prompt for the code generator in 14, and the user prompt for code generator in 15.

---

**System Prompt for Visual Critic.**

Task:
You are an advanced image analysis assistant. Multiple images will be provided along with their color statistics. The first image is the source image, and the rest of the images are the target images. The content and the photometric style of the source and target images differ. The photometric styles of all the target images are the same. Your task is to compare the source and target images in terms of the photometric style and identify how the target images differ from the source image in the specific photometric aspects: Exposure, Contrast, Highlight, Shadow, Saturation, Temperature, Texture.

Definition:
- Exposure refers to the overall brightness of the image. A higher factor results in a brighter image, while a lower factor makes the image darker.
- Contrast refers to the difference in brightness between light and dark areas of an image. A higher factor increases the difference, making the image more vivid but losing detail, while a lower factor reduces the difference, retaining more detail but making the image look softer.
- Highlight refers to the brightest areas in an image. A higher factor brightens these regions further, which can lead to loss of detail in overexposed areas, while a lower factor reduces brightness, helping to recover details lost in the highlights.
- Shadow refers to the darkest areas in an image. A higher factor brightens these regions, revealing details hidden in underexposed areas, while a lower factor darkens the shadows, enhancing contrast and creating a more dramatic effect, which can result in a loss of detail in the darkest areas.
- Saturation refers to the intensity of colors in an image. A higher factor enhances the vibrancy of colors, making them more intense, while a lower factor reduces the intensity, eventually leading to a grayscale image, where all color is removed.
- Temperature refers to the balance between warm and cool tones in an image. A higher factor adds warmth with reddish tones, while a lower factor introduces coolness with bluish tones.
- Texture refers to the level of detail and high-frequency variations in an image, influencing its perceived sharpness and surface characteristics. A higher factor enhances fine details and edges, while a lower factor softens the image by reducing these variations.

Instructions:
1. Choose whether to increase, decrease, or maintain the factor for each aspect. If adjusting, select the appropriate adjustment range from the given options, and if maintaining, write 'N/A' for that aspect.
2. If adjustments are needed for one or more aspects, write 'go' for the Overall part, while no adjustments are needed for any aspect, write 'stop'.

Output Format:
- Exposure: [description of exposure difference, e.g., the brightness of the target image is 10-20% higher than the one of the source image. or N/A.]
- Contrast: [description of contrast difference, e.g., the contrast of the target image is 10-20% higher than the one of the source image. or N/A.]
- Highlight: [description of highlight difference, e.g., the highlight of the target image is 10-20% higher than the one of the source image. or N/A.]
- Shadow: [description of shadow difference, e.g., the shadow of the target image is 10-20% higher than the one of the source image. or N/A.]
- Saturation: [description of saturation difference, e.g., the saturation of the target image is 10-20% higher than the one of the source image. or N/A.]
- Temperature: [description of temperature difference, e.g., the temperature of the target image

---

is 10-20% higher than the one of the source image. or N/A.]
- Texture: [description of texture difference, e.g., the texture of the target image is 10-20% higher than the one of the source image. or N/A.]
- Overall: Write 'Stop' if there is an N/A for all aspects, and 'Go' if one or more aspects have differences.

---

**User Prompt for Visual Critic.**

Task:
You should describe the similar parts between the source image and the target images and generate 3 candidate descriptions. Each candidate should include the difference of all the aspects. Compare the source image and the target images in terms of the photometric adjustments made to the image, and describe the difference in each aspect. You can choose the range from the following list: {range_list}%. Do not exceed the range. You can use the color statistics and the scores between the source and target image as a guide.

Color Statistics:
- Source: {source image statistics}.
- Targets (averaged): {average of target images statistics}.

Averaged scores (PSNR, SSIM, LPIPS, Delta E):
{Scores between source and reference images}

Output Format:
Similar parts
[Description of the similar parts]

Candidate 1
[Description of the first candidate]

Candidate 2
[Description of the second candidate]

Candidate 3
[Description of the third candidate]

---

**System Prompt for Code Generator.**

Task:
You are an expert Python programmer. Your task is to generate Python code that sets the appropriate filters and parameter values based on the given photometric aspect-wise description of the color tone difference between the source image and the target image, and arranges the sequence of those steps to make the source image resemble the target image.

Based on the given description, choose one of the following three options and proceed with the corresponding photometric adjustments:
- Global Brightness Adjustment (exposure, contrast): If global brightness adjustments are needed more than 1%, focus on modifying elements that affect overall brightness. Do not adjust local brightness, color tone, and texture elements at this stage, only global brightness-related factors.
- Local Brightness Adjustment (highlight, shadow): If the global brightness adjustments are completed with less than 1% differences, focus on modifying elements that affect local brightness. Do not adjust global brightness, color tone, and texture elements at this stage, only local brightness-related factors.
- Color Tone and Texture Adjustment (saturation, temperature, texture): If both the global and local brightness adjustments are completed with less than 1% differences, focus on

modifying elements that affect color tone and texture. Do not adjust global brightness and local brightness elements at this stage, only color tone and texture-related factors.

---

### Code Generation Instructions

Instructions:
1. Examine the given photometric difference description to determine which option to choose, and select only one option from the three options. Ensure that no other options are executed in the code.
2. Select the appropriate filters for the selected adjustment option, and arrange filters in the correct order.
3. The filter parameters can be chosen randomly within the range specified in the description.
4. The variable name of the adjusted image is "`{save_adj_img_name}`".

Difference Description:
`{Difference description from Visual Critic}.`

Available Functions:
- "filter.exposure(f_exp: float) -> np.ndarray": Adjusts the exposure (overall brightness) of an image. The f_exp parameter is an exposure adjustment factor, ranging from -1 to 1. The positive f_exp values brighten the overall image, while negative values darken it.
- "filter.contrast(f_cont: float) -> np.ndarray": Adjusts the contrast of an image by scaling its pixel values relative to the mean brightness of the image. The f_cont parameter is a contrast adjustment factor, ranging from -1 to 1. Positive f_cont values increase the contrast, making the image more vivid but potentially losing detail in bright and dark areas, while negative values reduce the contrast, retaining more detail but making the image look softer.
- "filter.highlight(f_high: float) -> np.ndarray": Adjusts the brightness of the bright areas of an image. The f_high parameter is a highlight adjustment factor, ranging from -1 to 1. The positive f_high values intensify the highlights, and negative values reduce them to recover details.
- "filter.shadow(f_shad: float) -> np.ndarray": Adjusts the brightness of the dark areas of an image. The f_shad parameter is a shadow adjustment factor, ranging from -1 to 1. The positive f_shad values brighten the shadows and negative values deepen them.
- "filter.saturation(f_sat: float) -> np.ndarray": Adjusts the saturation of an image. The f_sat parameter is a saturation adjustment factor, ranging from -1 to 1. The positive f_sat values increase color vibrancy, while negative values desaturate the image towards grayscale.
- "filter.temperature(f_temp: float) -> np.ndarray": Adjusts the color temperature of an image by modifying the balance between warm and cool tones in the RGB color space. The f_temp parameter is a temperature adjustment factor, ranging from -1 to 1. The positive f_temp values shift colors toward warmer tones by increasing red, while negative values shift colors toward cooler tones by enhancing blue.
- "filter.texture(f_text: float) -> np.ndarray": Adjusts the texture of an image by modifying its high-frequency details using Gaussian blur. The f_text parameter is a texture adjustment parameter, ranging from -1 to 1. The positive f_text values enhance texture by amplifying high-frequency details, while negative values soften texture.

Please return the code directly without any imports or additional explanations.
Ensure the code is clear, correct, and follows the steps logically.

---

### B.3 SELECTION SCORE

At each iteration, we generate three candidate images and select one as the source image for the next iteration. For the selection process, we employ a CLIP (Radford et al., 2021)-based scoring method. Specifically, we compute the probabilities of alignment between each image in the candidate set and the reference image with respect to eight textual prompts from four global filters: "a dark light photo" and "a bright light photo" from the exposure, "a low-contrast photo" and "a high-contrast photo" from the contrast, "a desaturated colours photo"

and "`a vivid colours photo`" from the saturation, and "`a cool-toned photo`" and "`a warm-toned photo`" from the temperature. We then calculate the KL Divergence between the probability distribution of each candidate image and that of the reference image. The image with the lowest score is selected. If multiple reference images are provided, we average their probability distributions before computing the error.

While we use CLIP-based similarity as selection scores in our experiments, exploring more sophisticated or perceptually aligned scoring metrics remains an open direction. For example, learning a task-specific scoring model may improve candidate selection and overall retouching quality. Developing an adaptive selection criterion that better aligns with user preferences or aesthetic judgments could further enhance the robustness and flexibility of the system.

### B.4 REFERENCE IMAGE SET CONSTRUCTION

We construct source–reference image pairs that simulate realistic user behavior in image retouching. In practice, users often choose reference images with semantically similar content. To mimic this behavior, we use CLIP (Radford et al., 2021) to extract image-level embeddings and compute pairwise KL divergence across the dataset. Reference images are selected based on similarity in the embedding space. Examples of such source–reference pairs are shown in Fig. 23.

## C  DETAILS OF EXPERIMENTS

### C.1  BRIGHTNESS RANGE PREDICTION TEST

In Table 1 of the main paper, the vision language model (VLM) with GPT-5 (OpenAI, 2025) is evaluated on its ability to infer the brightness adjustment range given a pair of images: the original and a manually brightened version. The task is framed as a classification problem over six predefined discrete intervals: (0–5), (5–10), (10–20), (20–40), (40–60), and (60–100). Given this setup, the accuracy of random guessing is approximately 16.7%. The prompt used for VLM can be found in 24.

---

**VLM prompt for range prediction test**

**System Prompt:** You are an image comparison model. Given two images, determine the brightness difference between them and choose the appropriate difference range from the following list: [(0,5), (5,10), (10,20), (20,40), (40,60), (60,100)]. For example, if the brightness difference is approximately 15%, respond with "(10,20)". Do not provide any additional explanations or details.
**User Prompt:** (Single) Choose the two most appropriate brightness difference range between the two images. (Multi) Choose the appropriate brightness difference range between the two images. The pixel means of the first image is `{The mean pixel value of the original image}` and the second image is `{The mean pixel value of the manually brightened image}`.

---

### C.2  FINE-TUNING SUPERVISED MODELS

To provide a fair and comprehensive comparison, we fine-tune the RSFNet (Ouyang et al., 2023) and PG-IA-NILUT (Kosugi, 2024) models on the same set of reference images used in our training-free photo retouching pipeline. Importantly, the supervised baselines are not trained from scratch using only five examples. Instead, they are fully trained in the standard manner on other styles, and the five target-style images are used solely for adaptation. For the MIT-Adobe FiveK evaluation, we initialize the baselines from weights trained on the PPRK10K dataset and adapt them using the five reference images from the target style of MIT-Adobe FiveK; conversely, for the PPRK10K evaluation, we start from weights trained on MIT-Adobe FiveK and adapt using the five reference images from the target style of PPRK10K. The results are shown in Table 10, where we look at five different styles for MIT-Adobe FiveK and three different styles for PPRK10K.

We use the exact same training parameters provided by the authors' code. For PG-IA-NILUT this involves three training stages and for RSFNet this involves just a single training stage. Due to the

smaller training set, we determined the number of training iterations through experimentation to avoid overfitting. For RSFNet, we used 100 iterations, and for PG-IA-NILUT we used 200 iterations. Training for longer resulted in degraded performance. However, despite this advantage, our training-free approach is still competitive or even outperforms both of these fine-tuned baselines across all evaluation metrics.

### C.3 OTHER SELECTION SCORES

In Table 6 in the main text, we compare our selection score with alternative methods: (1) RGB-channel histograms, (2) YUV-channel histograms, (3) Gram matrix similarity commonly used in style transfer, (4) our KL CLIP score using prompts regarding all filters including local ones, and (5) our default KL CLIP score using only global filters.

For (1) RGB-channel and (2) YUV-channel histogram-based scores, we compute the channel-wise histogram matching loss $L$ between the images from the candidate set $x^i \in \mathcal{C}$ and the reference images $\mathcal{Y} = \{y^j\}_{j=1}^M$, and add all of them to get the distance

$$D_i^{RGB} = \frac{1}{M} \sum_{j=1}^M \left( L_{ij}^R + L_{ij}^G + L_{ij}^B \right), \quad D_i^{YUV} = \frac{1}{M} \sum_{j=1}^M \left( L_{ij}^Y + L_{ij}^{UV} \right). \tag{9}$$

For (3) gram matrix similarity, for each layer $\ell \in \mathcal{L}$, where $\mathcal{L}$ is a set of layers in VGG (Simonyan & Zisserman, 2015), the feature map of image $I$ is

$$F_\ell(I) \in \mathbb{R}^{C_\ell \times (H_\ell W_\ell)}, \qquad G_\ell(I) = \frac{1}{C_\ell H_\ell W_\ell} F_\ell(I) F_\ell(I)^\top. \tag{10}$$

The Gram-based style distance between source $x_i$ and target $y_j$ is

$$D_i^{\text{Gram}} = \frac{1}{M} \sum_{j=1}^M \sum_{\ell \in \mathcal{L}} \left\| G_\ell(x^i) - G_\ell(y^j) \right\|_F^2. \tag{11}$$

Finally, the candidate with the smallest distance is selected:

$$i^* = \arg\min_i D_i. \tag{12}$$

For (4), we additionally incorporate six prompts corresponding to three local filters, *e.g.*, highlight, shadow, and texture, when computing the CLIP alignment probabilities Radford et al. (2021). The prompts are: "`a photo with dim highlights`" and "`a photo with bright highlights`" for the highlight filter, "`a photo with dark shadows`" and "`a photo with bright shadows`" for the shadow filter, and "`a smooth photo`" and "`a sharp photo`" for the texture filter. All equations remain the same as in Sec. 3.1, except that the number of prompts $K$ is increased from 8 to 14.

### C.4 USER STUDY

In each question, users were shown five reference style images, one ground-truth image (for reference only), and four retouched results produced by different models. The order of the four candidate images was randomly shuffled to avoid positional bias. Participants were asked: "Which image best reflects the retouching style of the given reference images?" Each participant selected the single best result from the four options.

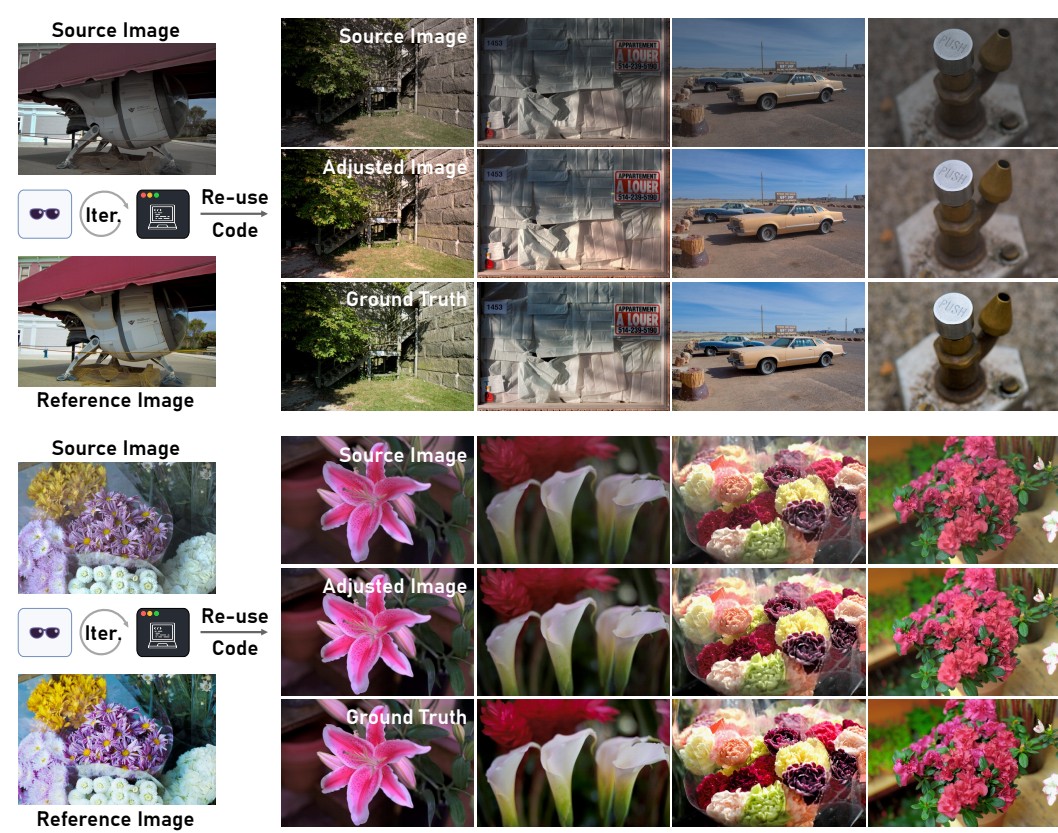

Figure 16: **Additional qualitative results for applying the restored filter, corresponding to Fig. 4 in the main paper.** The paired setup enables extracting a reusable retouching code that can be applied to other images like a preset filter. As shown, the code extracted from the left image pair can be reused to retouch other images, achieving a style similar to the GT without additional supervision.

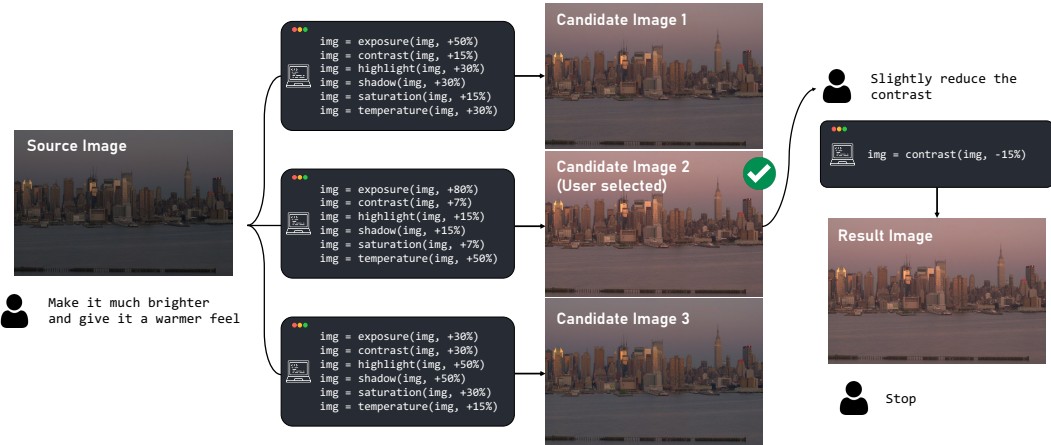

Figure 17: **User interactive retouching with user selection (qualitative process of user interaction corresponding to Fig. 5 in the main paper).** The user can provide natural language instructions to retouch images towards the desired style and select a preferred image among the adjusted candidates. At each iteration, RetouchLLM generates three new candidates, and the user selects one to proceed.

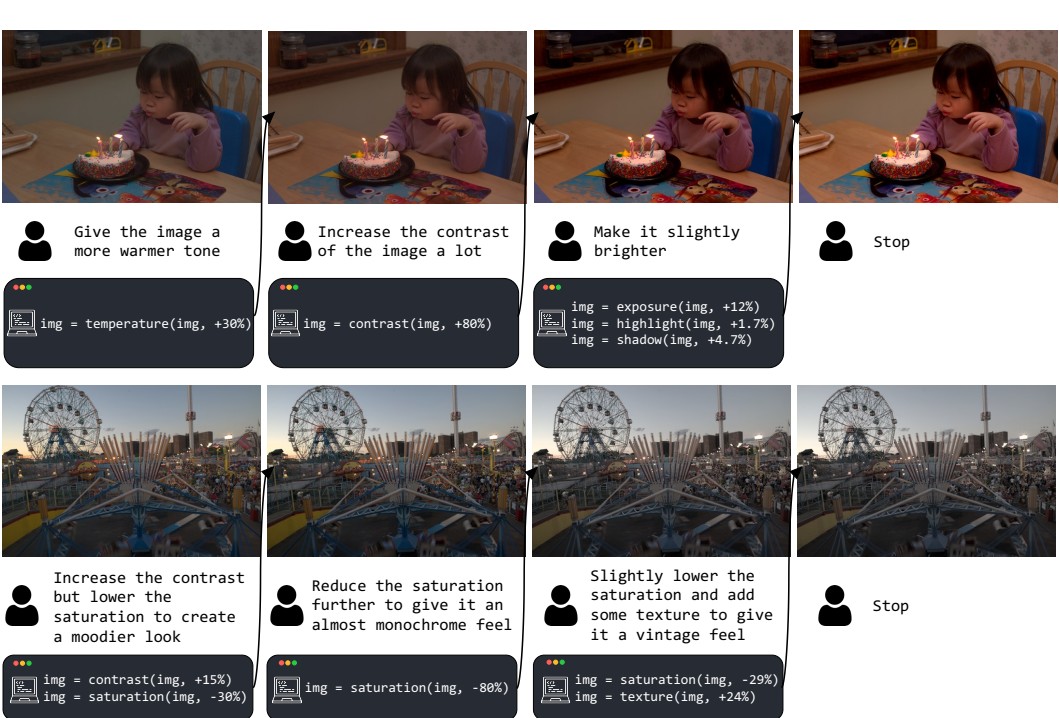

Figure 18: **Additional examples of user interactive retouching, corresponding to Fig. 5 in the main paper.** The user can provide instructions to retouch images towards the desired style.

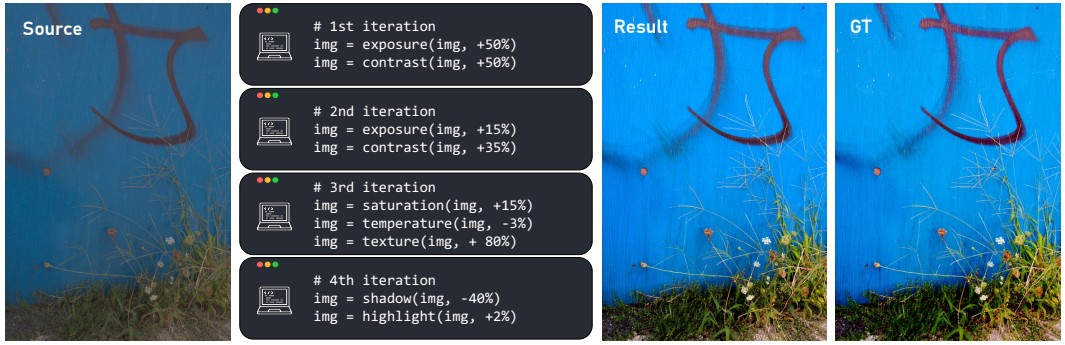

Figure 19: **Qualitative results for code generation.** The code generator plans the coarse-to-fine retouching process by starting with global adjustments and then focusing on the finer details.

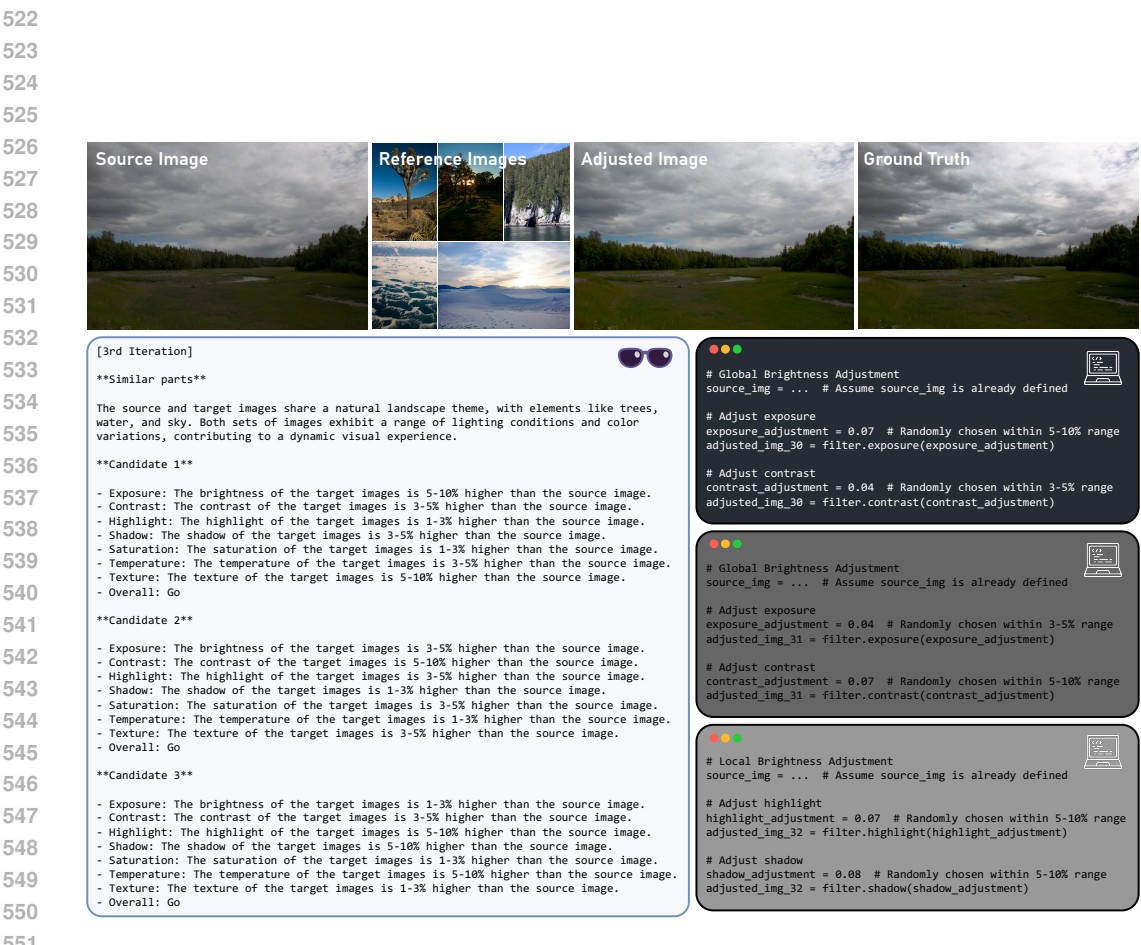

Figure 20: **Generated descriptions and corresponding codes samples 1**

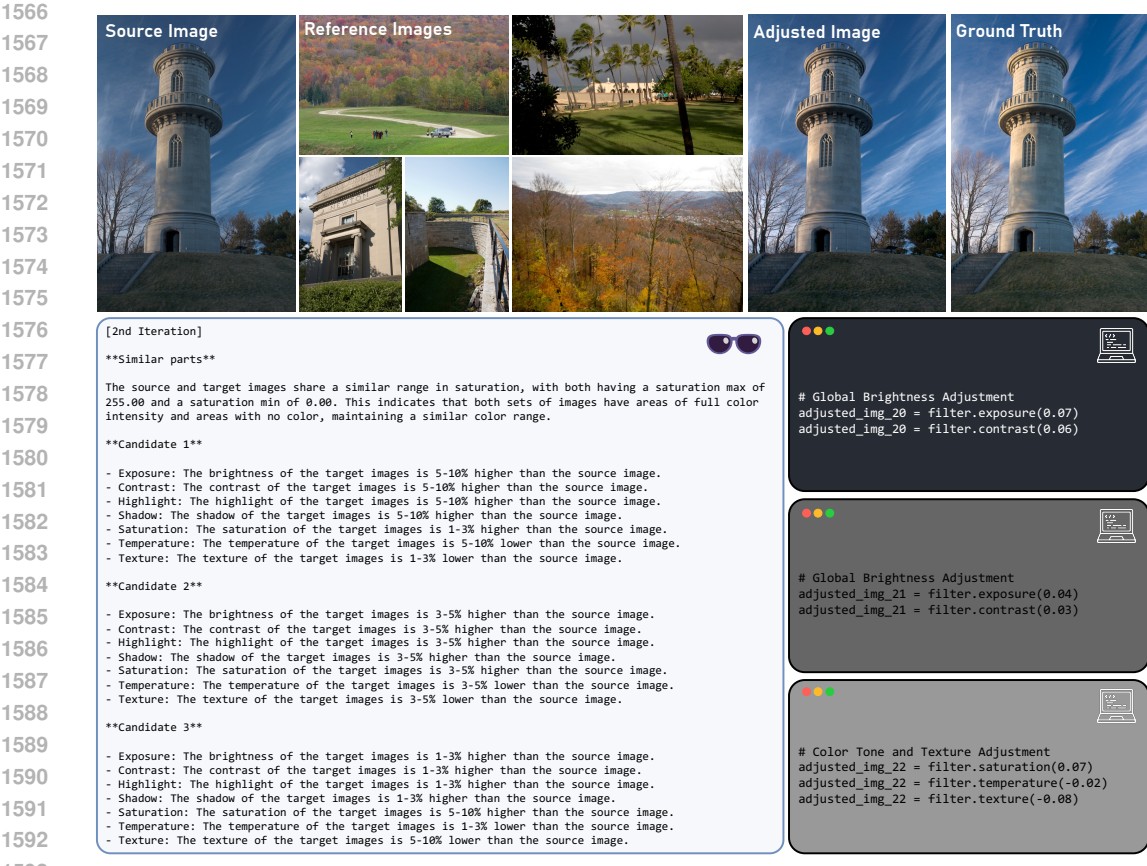

Figure 21: **Generated descriptions and corresponding codes samples 2**

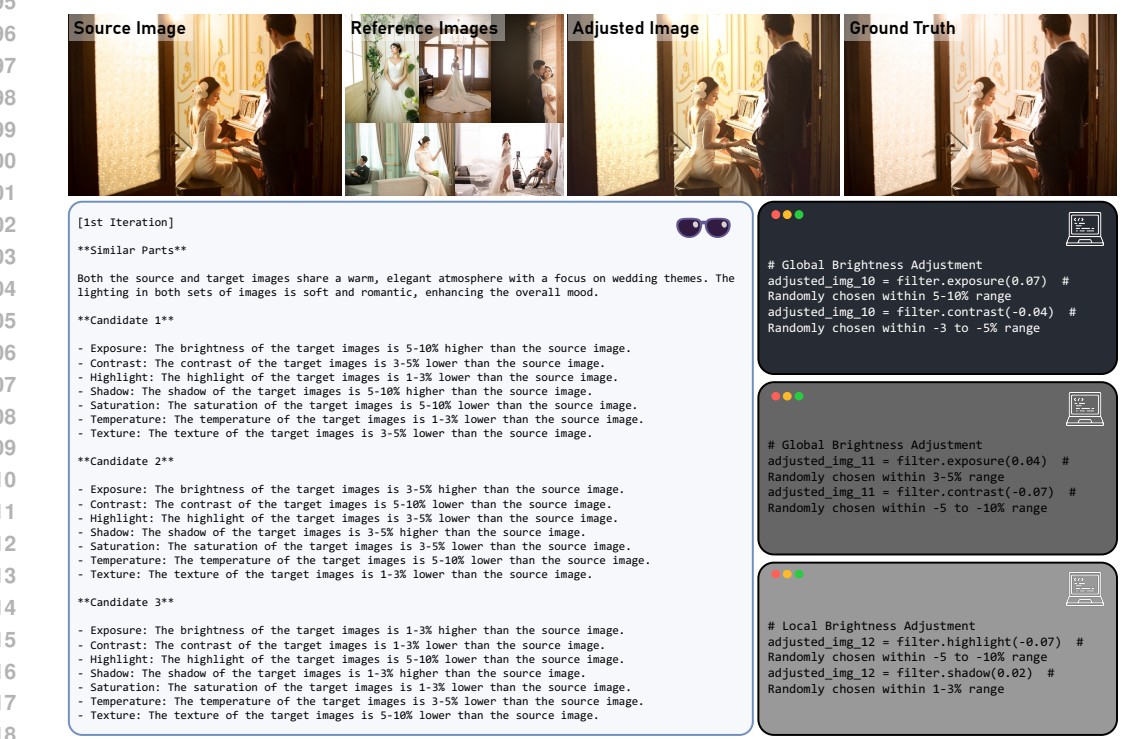

Figure 22: **Generated descriptions and corresponding codes samples 3**

**Source Image**                    **Reference Images**

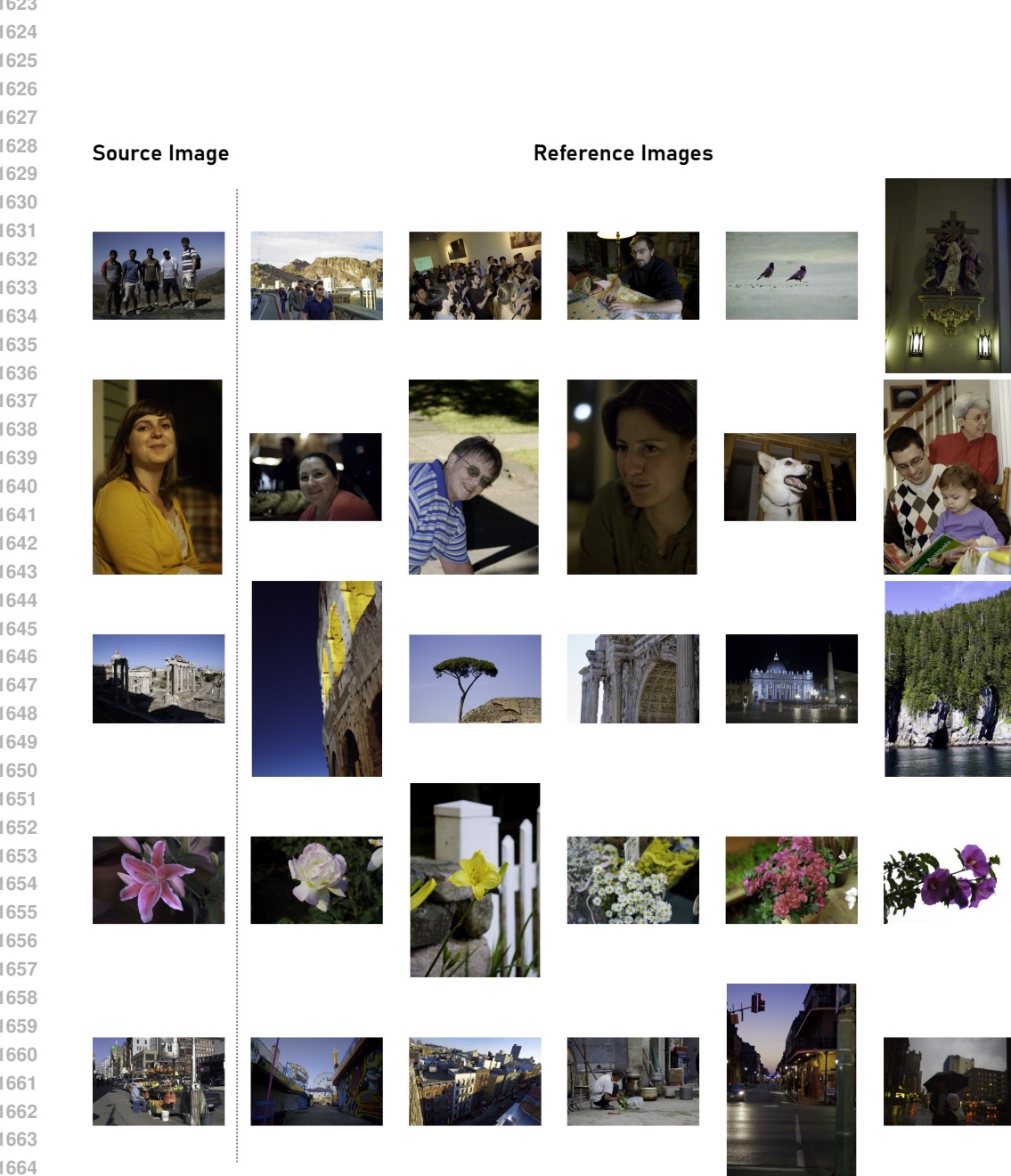

Figure 23: **Constructed reference image set examples from MIT-Adobe FiveK.**

