# OpenReview forum: "RetouchLLM: Training-free Code-based Image Retouching with Vision Language Models"
_ICLR.cc/2026/Conference — Submitted to ICLR 2026_

### Official Review · Reviewer_o8mv · 2025-10-31

**Soundness:** 3
**Presentation:** 3
**Contribution:** 1
**Rating:** 4
**Confidence:** 3

**Summary:**

The paper proposes a training-free workflow, RetouchLLM, for image retouching. RetouchLLM uses VLMs and LLMs to understand and suggest possible enhancement operations for the input images. Unlike training a deep neural network, It is training-free and white-box, no need for large-scale paired data. RetouchLLM achieves notable improvement over previous methods.

**Strengths:**

### Idea/Motivation
 - The core idea of RetouchLLM is to use powerful priors from LLMs/VLMs and their understanding ability to make the inference transparent.
- Instead of using an expensive training pipeline and black-box neural networks, a white-box and code-based workflow is proposed.

### Method
- Two core components, a visual critc and a code generator are proposed.
- The model uses an iterative manner with a CLIP scorer to make sure a stable convergence.
- With a code-based editing, the resolution of the input images are not lost.


### Experimental results
- Although training-free, RetouchLLM outperforms or achieves competitive results against baselines on MIT-Adobe 5k and PPR10K, against many trained models.
- The framework shows a good generalization ability with diverse styles and different LLMs/VLMs.

### Writing/Delivery
- The paper is easy-to-follow and overall well-structured.

**Weaknesses:**

### Idea/Motivation
- Although it is new to Image Retouching, using VLMs or LLMs for image editing is not very new. For example, Cropper [1] uses a similar idea for image cropping, with VLMs and introduces a training-free approach.

### Method
- Iterative refinement seems not a novel approach, as it is also shown in the Cropper paper.
- There is a compute overhead. RetouchLLM generates N candidates per iteration (up to T=10 in practice). Compared to other training-based methods, it might be more fair to report additional costs.

### Experimental results
- User study scale. The user study only investigates 25 participants. (L489)
- Few “negative examples.” The paper would be more comprehensive if it includes some failure analysis.


### References
[1] Lee S H, Jiang J, Xu Y, et al. Cropper: Vision-Language Model for Image Cropping through In-Context Learning[C]//Proceedings of the Computer Vision and Pattern Recognition Conference. 2025: 30010-30019.

**Questions:**

Please kindly refer to weaknesses.

---

> ### Author Response · Authors · 2025-11-21
>
> We gratefully thank the reviewer o8mv for recognizing the strengths of our work, including the **inference transparency** through **a white-box, code-based approach**; our design ensures **a stable convergence**; the ability to **preserve full resolution** through code-based editing; our **competitive performance** compared to training-based methods; the **solid generalization** across diverse styles and model choice; and the **clear organization of the paper**. We have addressed all concerns in detail and kindly invite the reviewer to re-evaluate our work based on our responses and the revised paper and Appendix, where all modifications are marked in blue for convenience.
>
> **W1 & W2. Clarifying the difference between Cropper and ours**
> Cropper differs fundamentally from our work in both the action space and the task objective. Cropper performs discrete spatial cropping by selecting a bounding box within the image, whereas our method performs continuous photometric retouching that adjusts tone, color, and style at the pixel level. This reflects a deeper distinction: spatial region selection relies on object-level semantics, while photometric retouching requires reasoning over global appearance and style consistency.
>
> In addition, the refinement procedure differs in its operational details: Cropper does not retain the original state and therefore performs only a shallow two-step adjustment, whereas our method keeps the source as a candidate at every step, allowing rollback and enabling a multi-step refinement process in a continuous photometric space.
>
> For this reason, the similarity in using VLMs/LLMs or an iterative paradigm does not imply that the novelty is diminished: the paradigm may be shared, but the task, action space, objectives, and required reasoning are entirely different. Our contribution lies in extending VLM/LLM capabilities to continuous, style-oriented retouching, which has not been addressed in prior training-free or VLM/LLM-based editing frameworks.
>
> **W3. Interpreting cost as test-time adaptation for training-free personalized retouching**
> A straightforward way to further reduce latency is to parallelize the candidate-wise code-generation steps, which are currently executed sequentially. Since processing a single candidate takes 84.32s, batching can reduce the overall latency to a similar time, which we observe to be 86.44s. In addition, using early stopping after five steps to balance quality and runtime further reduces latency to 54.45 seconds while incurring minimal degradation in PSNR.
>
> Training-based approaches are still faster, but they require pre-training on fixed distributions and cannot adapt to a new user style without additional data and retraining. In contrast, our method immediately aligns with arbitrary reference styles without any training cost. Therefore, the latency should be interpreted as test-time adaptation for training-free personalized retouching rather than as model inefficiency. Furthermore, standard acceleration techniques (e.g., model distillation, diffusion-based LLMs) could further shorten the runtime without changing the overall system.
> |  | Original implementation | w/ candidate batching | w/ early stopping |
> | :---: | :---: | :---: | :---: |
> | Test-time (s) | 117.31 | 86.44 | 54.54 |
>
> **W4. Scale of the user study**
> We expanded the user study to address this concern. In addition to the original 25 participants, we collected responses from 15 additional participants, resulting in a total of 40 users. Each participant evaluated 30 samples, yielding 1,200 responses overall. The results remain highly consistent with our original findings: NILUT received 16.42%, RSFNet 9.67%, Z-STAR 3.17%, and our method 70.75%, indicating that the user preferences are stable even with a larger participant pool. We have updated the main manuscript accordingly.
>
> **W5. Analyzing negative examples**
> We previously discussed failure cases that occur during the retouching process in Appendix A.6. In addition, we include the failure cases of our final results in Fig. 10, and provide further discussion in Appendix A.6. In summary, certain regions tend to be correctly adjusted while other areas become overly or insufficiently edited, depending on which part the model focuses on. Such issues are expected to be addressed once mask-based local editing is supported, as shown in Fig. 11\. In addition, there are cases where the overall tone appears slightly under-brightened or mildly cool. Although these results do not perfectly match the ground truth, they still fall within a plausible range of adjustments. For additional discussion and detailed examples, please refer to Appendix A.6.

---

### Official Review · Reviewer_gY3G · 2025-11-01

**Soundness:** 3
**Presentation:** 3
**Contribution:** 3
**Rating:** 6
**Confidence:** 3

**Summary:**

The paper proposes RetouchLLM, a training-free, white-box, code-driven image retouching framework that operates directly on high-resolution images. The system runs an iterative loop: a VLM describes stylistic gaps between a source image and a small set of style references; an LLM converts that description into an executable Python retouching program that composes a small pool of interpretable filters (exposure, contrast, saturation, temperature, highlight, shadow, texture). The same framework also supports a user-interactive mode.

**Strengths:**

- **Clear and simple approach**
The pipeline (critic→code generator→execute→select) is simple, elegant, and easy to extend (new filters or rules without retraining). The executable code improves the interpretability & reproducibility of the retouching process.

- **Training-free approach**
The approach doesn't need paired data or finetuning, directly handling high-res inputs. This dramatically lowers adoption barriers for photographers and apps.

- The paper provides a novel practice that replaces latent black boxes with explicit programs. This practice could inspire follow-up work beyond retouching.

**Weaknesses:**

- **Limited Evaluation**
The comparison includes Z-STAR and two supervised white-box/fine-tuning baselines, RSFNet, PG-IA-NILUT. But classic white-box pipelines (e.g., Exposure, Harmonizer-style operators, Neural Color Operators, LUT-based methods, or rule-based/optimization approaches) are encouraged to be included, as well as additional comparison with other LLM-based methods, such as MonetGPT.

- **Limited Novelty**
The conceptual novelty is somewhat incremental relative to the growing body of LLM/VLM tool-use and program-synthesis frameworks. The paper’s novelty lies in demonstrating that this approach is competitive for high-res retouching and yields reusable presets, which is not a fundamentally new pipeline.

**Questions:**

See weaknesses

---

> ### Author Response · Authors · 2025-11-21
>
> We gratefully thank the reviewer gY3G for highlighting the strengths of our work, including the **clarity and simplicity of our pipeline**, whose is **elegant, extensible, and fully interpretable** through executable code; the **training-free design** that operates directly on high-resolution inputs, thereby **lowering barriers for real-world use**; and the **novelty of replacing latent black-box mechanisms with explicit programs**, which **could inspire further research** beyond retouching. We have addressed all concerns in detail, and we kindly invite the reviewer to re-evaluate our work based on our responses and the revised paper and Appendix, where all modifications have been marked in blue for convenience.
>
> **W1. Comparison with more baselines**
> We compare against additional train-based models, including Exposure and MonetGPT, in the plausibility assessment setting \[Dutt et al. 2025\], where each original image has five expert retouched versions, and the model output is compared to whichever expert result is closest. This measures how plausibly the model can match human editing preferences. The table below shows that our approach still achieves strong performance, indicating that the edits produced by our model are not only quantitatively superior to other methods but also more aligned with the range of human expert adjustments. We have added these results in Sec. 4.2 (Table 4).
>
> | Method | PSNR (↑) | SSIM (↑) | LPIPS(↓) | ∆E(↓) |
> | :---- | :---: | :---: | :---: | :---: |
> | Exposure \[Hu et al. 2018\] | 15.12 | 0.63 | 0.14 | \- |
> | Unpaired \[Kosugi et al. 2020\] | 21.73 | 0.83 | 0.12 | \- |
> | RSFNet \[Ouyang et al. 2023\] | 21.85 | 0.88 | 0.08 | \- |
> | InstructP2P \[Brooks et al. 2023\] | 16.99 | 0.61 | 0.22 | \- |
> | MGIE \[Fu et al. 2024\] | 22.94 | 0.74 | 0.08 | \- |
> | MonetGPT \[Dutt et al. 2025\] | 23.75 | 0.90 | **0.07** | \- |
> | RetouchLLM (Ours) | **25.48** | **0.92** | 0.09 | **7.57** |
>
> **W2. Clarifying the novelty beyond existing tool-use paradigms**
> Our contribution does not lie in the tool-use paradigm itself, but in how we adapt it to the retouching setting. The key novelty is the design of a stable, iterative retouching procedure driven by our selection score, which enables reliable refinement rather than one-shot operator prediction. This stands in contrast to prior retouching methods that learn monolithic models to regress operator parameters; our explicit, code-based formulation makes the process interpretable and supports direct high-resolution retouching without additional training. As the Reviewer gY3G noted, this design also represents a meaningful step toward enabling retouching workflows that can flexibly adapt to diverse user styles, and further, toward moving beyond fixed benchmarks.
>
> Even if our approach falls within the broader tool-use paradigm, the specific tools employed, how they are composed, and how they are operationalized for this task are fundamentally different. We believe that the resulting contributions are unique to the retouching setting and should be assessed on their own merits rather than being reduced to the shared high-level paradigm.

---

### Official Review · Reviewer_gEM5 · 2025-11-04

**Soundness:** 3
**Presentation:** 3
**Contribution:** 3
**Rating:** 6
**Confidence:** 3

**Summary:**

This work introduces a non-training-based image editing method. It evaluates the image based on LLM and iteratively updates the image by generating code. Eventually, it ensures that the image and the reference image have consistency across multiple dimensions. The main advantages of this solution are: (1) No training required, as it utilizes the evaluation ability of VLM for image similarity. (2) Interpretable, because the image adjustment is done through generated code, making it easy to observe. (3) Compatible with high-resolution. I am not familiar with this field, but I think the shortcomings seem to be straightforward as well. Compared with the end-to-end pair-based learning methods, this method may have poorer performance in fine-grained adjustments. The process of adjusting specific areas (for instance, perhaps some people would like certain areas such as the face or hands to have higher brightness and a smoother texture) through VLM is more indirect compared to the end-to-end approach.

**Strengths:**

- The organization of the paper is very good and it is easy to follow.
- The design of the selection score is novel.
- Compared to the end-to-end approach, this method can fully leverage the capabilities of VLM and has a strong zero-shot learning ability.
- The experimental results are conclusive and the outcome seems very promising.
- The visualization of the paper is done well, I enjoy reading it.

**Weaknesses:**

- I'm not familiar with this field, but I doubt the applicability of this method for some specific features of local image editing. Does VLM only have good adaptability in certain aspects such as overall contrast and brightness adjustments?
- The baselines seem to be too few. As a method that doesn't require training, I think having more baselines is necessary.
- The choice of filters seems to be limited. Is this limitation due to interpretive requirements? If we abandon interpretability, for instance by using the prompt learning approach, would there be better results?

**Questions:**

- How applicable is the model? Is it only applicable to certain types of image editing on a global scale? However, I am not familiar with this field and I am unsure if this task actually only considers modifications on a global scale.
- How was the baseline selected? I think this is too little. Papers related to LLM should ensure that the baseline is adequately defined.
- How sensitive is the prompt? Does the prompt for visual critic seem to have been designed manually?

---

> ### Author Response · Authors · 2025-11-21
>
> We sincerely thank the reviewer gEM5 for recognizing the strengths of our work, including the **clear and well-structured** organization of the paper; the **novelty of our selection score design**; the **strong zero-shot performance** compared to end-to-end approaches; and the **conclusive and promising experimental results and visualizations**. We have addressed all concerns in detail, and we kindly invite the reviewer to re-evaluate our work based on our responses and the revised paper and Appendix, where all modifications have been marked in blue for convenience.
>
> **W1 (Q1). Applicability for local image editing**
> To evaluate whether our pipeline can be extended toward localized retouching, we additionally incorporated local editing using segmentation-based masking. Specifically, a target region mask was obtained via SAM and applied after the original global enhancement. The results in Fig. 11 (Appendix A.7) demonstrate that local editing further improves similarity to the ground truth compared to using global operations alone, e.g., the first sample's PSNR is increased from 23.79 to 29.33, indicating that the framework can be extended beyond purely global corrections by integrating mask-guided refinements.
>
> **W2 (Q2). Comparison with more baselines**
> We compare against additional train-based models in the plausibility assessment setting \[Dutt et al. 2025\], where each original image has five expert retouched versions, and the model output is compared to whichever expert result is closest. This measures how plausibly the model can match human editing preferences. The table below shows that our approach still achieves strong performance, indicating that the edits produced by our model are not only quantitatively superior to other methods but also more aligned with the range of human expert adjustments. We have added these results in Sec. 4.2 (Table 4).
> | Method | PSNR (↑) | SSIM (↑) | LPIPS(↓) | ∆E(↓) |
> | :---- | :---: | :---: | :---: | :---: |
> | Exposure \[Hu et al. 2018\] | 15.12 | 0.63 | 0.14 | \- |
> | Unpaired \[Kosugi et al. 2020\] | 21.73 | 0.83 | 0.12 | \- |
> | RSFNet \[Ouyang et al. 2023\] | 21.85 | 0.88 | 0.08 | \- |
> | InstructP2P \[Brooks et al. 2023\] | 16.99 | 0.61 | 0.22 | \- |
> | MGIE \[Fu et al. 2024\] | 22.94 | 0.74 | 0.08 | \- |
> | MonetGPT \[Dutt et al. 2025\] | 23.75 | 0.90 | **0.07** | \- |
> | RetouchLLM (Ours) | **25.48** | **0.92** | 0.09 | **7.57** |
>
> **W3. Trade-off between interpretable filter design and prompt learning**
> We adopt interpretable operators because prior work has consistently shown that explicit parameter prediction for well-defined editing operators yields greater perceptual stability and more reliable content preservation than direct image-to-image regression. In contrast, prompt learning methods fall into the same category as end-to-end regression approaches: they typically require large-scale paired datasets and can exhibit instability or content degradation when adapting to new styles. Our operator-based formulation is also fully extensible; additional filters can be incorporated simply by defining them as Python-style functional editing operators, enabling the system to expand its expressive capacity.
>
> **Q3. Model-agnostic prompting**
> The visual critic prompt is manually designed but intentionally kept model-agnostic. We use the same prompt for all architectures, and wording changes do not noticeably affect performance, indicating that the prompt is not highly sensitive.

---

### Official Review · Reviewer_prGR · 2025-11-05

**Soundness:** 2
**Presentation:** 3
**Contribution:** 2
**Rating:** 4
**Confidence:** 4

**Summary:**

The paper proposes RetouchLLM, a training-free framework for image retouching. It operates iteratively: a VLM (Visual Critic) identifies photometric differences between source and reference images, and an LLM (Code Generator) translates these differences into executable Python code using 7 predefined filters. A CLIP-based selection score guides the process by choosing the best candidate at each iteration. The approach aims to provide an interpretable and adaptable alternative to data-driven methods.

**Strengths:**

1.  **White-Box Interpretability:** The code-based approach provides full transparency. The generated Python programs are explicit, reproducible, and reusable as preset filters (Fig 4).
2.  **Training-Free Adaptability:** The system operates without training data and can adapt to new styles using very few reference images (M=5), addressing the data dependency of supervised approaches.
3.  **Robust Engineering:** The iterative design with multi-candidate generation (Sec 3.2) is a sound strategy to mitigate the inherent uncertainty in VLM assessments.

**Weaknesses:**

1.  **Severely Restricted Action Space:** The framework is limited to 7 global photometric operations (Sec 3.3). It fundamentally lacks the capability for spatially localized adjustments (e.g., masking, selective color grading, complex curves), which restricts the method's scope to basic global correction rather than comprehensive retouching.
2.  **Impractical Latency:** The iterative process (T=10) requires sequential inferences from large foundation models, resulting in high latency (≈2 minutes per image, App B.1). This is prohibitively slow for practical use compared to feed-forward models.
3.  **Constrained Heuristic Search, Not Planning:** The claim of "Photo adjustment planning" is overstated. The LLM is explicitly forced by the system prompt (App B.2) to follow a rigid, predefined adjustment order (Global -> Local -> Color/Texture). This artificial constraint reduces the process to a heuristic search rather than genuine planning and prevents the exploration of optimal sequences.
4.  **Fragile Style Generalization:** Performance is bottlenecked by the VLM's ability to perceive style across different content. The significant performance drop between the paired (PSNR 29.21) and the more realistic unpaired (PSNR 22.19) setups (Table 4) highlights this fragility.
5.  **Insufficient Evaluation Protocol:** Comparisons with supervised models involve fine-tuning them on only 5 examples. This only demonstrates superiority in an extreme few-shot scenario and is insufficient to evaluate competitiveness against properly trained models.

**Questions:**

1.  **Localized Adjustments:** A significant limitation is the framework's reliance on global filters, which precludes spatially localized edits. This inability to perform region-specific adjustments (e.g., via spatial masks or curves) is a severe drawback. The authors should discuss potential extensions to incorporate spatially-varying operators to address this limitation.
2.  **Optimization Constraints:** Why was the rigid adjustment sequence (Global -> Local -> Color) enforced via prompting (W3)? How does performance change if the LLM is allowed to freely sequence any filter at any iteration?
3.  **Latency:** Are there concrete strategies to reduce the inference time (≈2 minutes) to practical levels?

---

> ### Author Response · Authors · 2025-11-21
>
> We sincerely thank the reviewer prGR for highlighting the strengths of our work, including the **white-box interpretability** enabled by our transparent, explicit, and reusable Python code; the **training-free adaptability** that allows our system to generalize to new styles; and the **robust engineering strategy** of iterative multi-candidate generation that **mitigates the inherent uncertainty in VLM assessments**. We have addressed all concerns in detail, and we kindly invite the reviewer to re-evaluate our work based on our responses and the revised paper and Appendix, where all modifications are marked in blue for convenience.
>
> **W1 (Q1). Extending the action space to include localized adjustments**
> To evaluate whether the restricted action space can be extended to include spatially localized retouching, we introduce local editing using segmentation-based masking. Specifically, a target region mask is obtained using SAM and then applied after the global enhancement. The result in Fig. 11 (Appendix A.7) demonstrates that local editing additionally improves similarity to the ground truth compared to using global operations alone, e.g., the first sample's PSNR is increased from 23.79 to 29.33, indicating that the framework can easily be extended beyond purely global adjustments.
>
> **W2 (Q3). Latency as test-time adaptation for training-free personalized retouching**
> A straightforward way to further reduce latency is to parallelize the candidate-wise code-generation steps, which are currently executed sequentially. Since processing a single candidate takes 84.32s, batching can reduce the overall latency to a similar time, which we observe to be 86.44s. In addition, using early stopping after five steps to balance quality and runtime further reduces latency to 54.45 seconds while incurring minimal degradation in PSNR.
>
> Feed-forward training-based approaches are still faster, but they require pre-training on fixed distributions and cannot adapt to a new user style without additional data and retraining. In contrast, our method immediately aligns with arbitrary reference styles without any training cost. Therefore, the latency should be interpreted as test-time adaptation for training-free personalized retouching rather than as a model inefficiency. Furthermore, standard acceleration techniques (e.g., model distillation, diffusion-based LLMs) could further shorten the runtime without changing the overall system.
> |  | Original implementation | w/ candidate batching | w/ early stopping |
> | :---: | :---: | :---: | :---: |
> | Test-time (s) | 117.31 | 86.44 | 54.54 |
>
> **W3 (Q2). Curriculum prior for retouch planning**
> Our prompt follows a common workflow used by professional retouchers, who typically apply global adjustments before editing local areas (also noted by Li et al., 2024). However, in our system, this serves only as a curriculum prior rather than a rigid rule. The actual sequence is decided dynamically: at each iteration, the code generator may return to global edits even if the previous step was local, or may combine operations based on the visual critic response. Thus, the system performs dynamic refinement at each iteration rather than a rigid heuristic search.
>
> To investigate the effect of the curriculum prior prompt, we compare performance with and without the corresponding phrase, as shown in the table below. When the prompt does not specify the ordering, the performance slightly degrades compared to the original setting, but it still remains superior to other competing models.
> | Method | PSNR (↑) | SSIM (↑) | LPIPS(↓) | ∆E(↓) |
> | :---: | :---: | :---: | :---: | :---: |
> | RSFNet | 18.03 | 0.773 | 0.178 | 18.34 |
> | PG-IA-NILUT | 20.54 | 0.743 | 0.168 | 12.13 |
> | Z-STAR | 16.01 | 0.607 | 0.397 | 17.70 |
> | RetouchLLM w/o prior | 21.75 | 0.902 | 0.082 | 11.15 |
> | RetouchLLM w/ prior | 22.19 | 0.909 | 0.070 | 10.07 |
>
> Li et al., Real-Time Exposure Correction via Collaborative Transformations and Adaptive Sampling. CVPR, 2024\.

---

> ### Author Response · Authors · 2025-11-21
>
> **W4. Style generalization under similar content conditions**
> As noted in L287–289, users generally rely on reference images with similar content. Following this practical assumption, we evaluated our method using different but content-aligned reference images, running the experiment seven times and reporting the mean and standard deviation (mean ± std) across five repetitions, excluding the maximum and minimum values. The result table below shows that our method behaves reliably when the content gap is moderate.
>
> When the reference images contain very different content, the performance may decrease because current VLMs may struggle to perceive and extract a consistent style across such heterogeneous scenes. We have added a discussion of this limitation, together with the repeating experiment results under similar content, in Sec. 4.2 (Table 9). Note that, as stated in L455-457, the PSNR value of the paired setup corresponds to the upper bound obtained by using the ground truth image as the reference, and is therefore not directly comparable to the unpaired setting.
> | PSNR (↑) | SSIM (↑) | LPIPS(↓) | ∆E(↓) |
> | :---: | :---: | :---: | :---: |
> | 22.34 ± 1.55 | 0.918 ± 0.016 | 0.068 ± 0.011 | 9.44 ± 1.04 |
>
> **W5. Clarifying the few-shot adaptation setting in supervised comparisons**
> The supervised baselines are not trained from scratch using only five examples. Rather, they are fully trained in the standard manner on other styles, and the five target style images are used only for adaptation (i.e., a style-specific fine-tuning stage), which is a common and fair comparison setting for style-guided retouching models. We have clarified this distinction and added the corresponding details in Appendix C.2.
>
> We also compare against additional train-based models in the plausibility assessment setting \[Dutt et al. 2025\], where each original image has five expert retouched versions, and the model output is compared to whichever expert result is closest. This measures how plausibly the model can match human editing preferences. The table below shows that our approach significantly outperforms all other models, indicating that the edits produced by our model are not only quantitatively superior to other methods but also more aligned with the range of human expert adjustments. We have added these results in Sec. 4.2  (Table 4).
> | Method | PSNR (↑) | SSIM (↑) | LPIPS(↓) | ∆E(↓) |
> | :---- | :---: | :---: | :---: | :---: |
> | Exposure \[Hu et al. 2018\] | 15.12 | 0.63 | 0.14 | \- |
> | Unpaired \[Kosugi et al. 2020\] | 21.73 | 0.83 | 0.12 | \- |
> | RSFNet \[Ouyang et al. 2023\] | 21.85 | 0.88 | 0.08 | \- |
> | InstructP2P \[Brooks et al. 2023\] | 16.99 | 0.61 | 0.22 | \- |
> | MGIE \[Fu et al. 2024\] | 22.94 | 0.74 | 0.08 | \- |
> | MonetGPT \[Dutt et al. 2025\] | 23.75 | 0.90 | **0.07** | \- |
> | RetouchLLM (Ours) | **25.48** | **0.92** | 0.09 | **7.57** |

---

### Author Response · Authors · 2025-11-21
**Summary of rebuttal**

Dear Area Chair and Reviewers,

We would like to briefly summarize the main points of our rebuttal. We have revised the paper and the Appendix, where all modifications are marked in blue for convenience.

1) **Better performance against additional baselines**: Reviewers prGR, gEM5, and gY3G request more baselines for comparison. We compare against additional training-based models in the plausibility assessment setting, and the results show that our method still outperforms these additional baselines and is also more aligned with the range of human expert adjustments.
| Method | PSNR (↑) | SSIM (↑) | LPIPS(↓) | ∆E(↓) |
| :---- | :---: | :---: | :---: | :---: |
| Exposure \[Hu et al. 2018\] | 15.12 | 0.63 | 0.14 | \- |
| Unpaired \[Kosugi et al. 2020\] | 21.73 | 0.83 | 0.12 | \- |
| RSFNet \[Ouyang et al. 2023\] | 21.85 | 0.88 | 0.08 | \- |
| InstructP2P \[Brooks et al. 2023\] | 16.99 | 0.61 | 0.22 | \- |
| MGIE \[Fu et al. 2024\] | 22.94 | 0.74 | 0.08 | \- |
| MonetGPT \[Dutt et al. 2025\] | 23.75 | 0.90 | **0.07** | \- |
| RetouchLLM (Ours) | **25.48** | **0.92** | 0.09 | **7.57** |

2) **Extending to local image editing**: Reviewers prGR and gEM5 are concerned about local editing. We provide examples (Figure 11 in Appendix A.7) that additionally incorporate local editing via segmentation-based masking. The results show that mask-based local editing further improves similarity to the ground truth (e.g., from 23.79 to 29.33 in PSNR), indicating that the framework can be extended beyond purely global corrections by integrating mask-guided refinements.
3) **On test-time cost**: Reviewers prGR and o8mv are concerned about the test-time cost. We reduce inference time via candidate batch processing and early stopping, reducing it from 117.31 sec to 54.45 sec. Although this remains slower than feed-forward training-based methods, our method should be interpreted as test-time adaptation for training-free personalized retouching, while also achieving much better photometric retouching results. The latency can be further reduced with additional acceleration engineering.
|  | Original implementation | w/ candidate batching | w/ early stopping |
| :---: | :---: | :---: | :---: |
| Test-time (s) | 117.31 | 86.44 | 54.54 |

4) **On novelty**: While reviewers gY3G and o8mv acknowledge the strengths of our method, including its training-free full-resolution pipeline that lowers barriers for real-world use and its replacement of latent black-box mechanisms with explicit executable programs, they express concerns that our approach may not be novel due to its use of a tool-based or iterative paradigm. We clarify that although the paradigm may be shared, the task, action space, objectives, and required reasoning are fundamentally different. Our work extends VLM and LLM reasoning to continuous, style-oriented retouching, which requires fine-grained photometric regulation rather than the discrete, object-focused paradigms explored in prior tool-based editing. Furthermore, our candidate generation and selection criteria are a novel component that is specific to this retouching task, which also brings significant improvements.

---

### Meta-Review · Area_Chair_uyh5 · 2026-01-07

**Summary:**

This paper introduces RetouchLLM, a training-free, white-box image retouching system that performs interpretable, code-based edits on high-resolution images without requiring paired training data. The method enables diverse, user-controllable retouching through multi-step adjustments guided by visual feedback and natural-language interaction.

Overall, the reviews are split. The key concerns about technical novelty and limited applications are raised and shared by multiple reviewers. As these major concerns are not convincingly addressed, the AC finds no ground to accept.

**Reviewer Concerns:**

Reviewers prGR, gEM5, and gY3G request more baselines for comparison:
The rebuttal provides a new table comparing additional training-based methods with the plausibility assessment setting.

Reviewer prGR: "Severely Restricted Action Space"
The rebuttal provides preliminary results of local editing (The result in Fig. 11 in Appendix A.7)
The AC checked the figure. While the PSNR is higher than the global edited results, the visual difference is quite limited. It looks simple mask-based intensity adjustment. The AC considers the concerns about the constraints of the limited action space to be still outstanding.

Reviewer gEM5: Comparison with more baselines
The rebuttal provided several learning-based baselines.

Reviewer gY3G: "The conceptual novelty is somewhat incremental relative to the growing body of LLM/VLM tool-use and program-synthesis frameworks."
The rebuttal clarified novelty beyond existing tool-use paradigms

Reviewer o8mv: "limited novelty wrt VLMs/LLMs or an iterative paradigm"

**Reviewer Scores:**

Reviewer prGR: 4: marginally below the acceptance threshold

Reviewer gEM5: 6: marginally above the acceptance threshold.

Reviewer gY3G  6: marginally above the acceptance threshold.

Reviewer o8mv: 4: marginally below the acceptance threshold.

Unfortunately, reviewers do not participate in the discussions. The AC checked the reviews and the rebuttals. Two key concerns remain:

1) technical novelty in iterative VLM for image editing:
The rebuttal argues that the differences in action space and retouching applications. But simply applying existing techniques to new applications does not imply sufficient technical contributions.

2) limited action space. This prevents this framework from performing more complicated retouching operations.

Overall, the AC believes that the reviewers would not have changed their scores if they participate fully in the dicsussions.

---

### Decision · Program_Chairs · 2026-01-26

Reject